# Tuning the role of charge-transfer states in intramolecular singlet exciton fission through side-group engineering

Steven Lukman[1], Kai Chen[2], Justin M. Hodgkiss[2], David H. P. Turban[1], Nicholas D. M. Hine[3], Shaoqiang Dong[4], Jishan Wu[4], Neil C. Greenham[1] & Andrew J. Musser[1,†]

Understanding the mechanism of singlet exciton fission, in which a singlet exciton separates into a pair of triplet excitons, is crucial to the development of new chromophores for efficient fission-sensitized solar cells. The challenge of controlling molecular packing and energy levels in the solid state precludes clear determination of the singlet fission pathway. Here, we circumvent this difficulty by utilizing covalent dimers of pentacene with two types of side groups. We report rapid and efficient intramolecular singlet fission in both molecules, in one case via a virtual charge-transfer state and in the other via a distinct charge-transfer intermediate. The singlet fission pathway is governed by the energy gap between singlet and charge-transfer states, which change dynamically with molecular geometry but are primarily set by the side group. These results clearly establish the role of charge-transfer states in singlet fission and highlight the importance of solubilizing groups to optimize excited-state photophysics.

[1] Cavendish Laboratory, J. J. Thomson Avenue, University of Cambridge, Cambridge CB3 0HE, UK. [2] MacDiarmid Institute for Advanced Materials and Nanotechnology, and School of Chemical and Physical Sciences, Victoria University of Wellington, Wellington 6010, New Zealand. [3] Department of Physics, University of Warwick, Gibbet Hill Road, Coventry CV4 7AL, UK. [4] Department of Chemistry, National University of Singapore, 3 Science Drive 3, Singapore 117543, Singapore. † Present address: Department of Physics and Astronomy, University of Sheffield, Sheffield S3 7RH, UK. Correspondence and requests for materials should be addressed to A.J.M. (email: a.musser@sheffield.ac.uk).

Singlet exciton fission (SF) is a photophysical process unique to organic semiconductors in which a singlet excited state separates into a pair of spin-triplet states. Due to the spin correlation of the product triplets, SF entails no change in overall spin and can thus proceed efficiently and on ultrafast timescales. Interest has recently increased due to the potential of such exciton multiplication to enhance solar cell efficiencies, with demonstrations of quantum efficiencies >100% (refs 1,2). However, the family of SF candidate materials remains relatively small, with most focus on pentacene, tetracene and 1,3-diphenyliso-benzofuran, and their derivatives[3–5]. To expand the pool of suitable chromophores, it is necessary to better understand the underlying mechanism of triplet formation. This is complicated in typical solid-state materials, where morphology and crystal packing can play a significant role and are difficult to control[6,7].

To circumvent these difficulties, recent efforts have focused on intramolecular SF in covalent dimers, most notably in various dimers of pentacene[8–12]. These materials provide uniquely detailed insight into the SF mechanism, particularly with regard to the influence of molecular geometry and charge-transfer (CT) states. Many leading theoretical models of SF assign a crucial mediating role to CT states[13–15]. These are generally calculated to exhibit strong coupling to singlet and triplet-pair states, even when they are energetically inaccessible, and could drive efficient triplet formation. These CT states are difficult to observe directly, but evidence of solvent-polarity-dependent SF rates in two classes of pentacene dimer[8,11,12] and dimers of

1,3-diphenylisobenzofuran[16] supports such a model. Likewise, a recent report of terrylenediimide dimers[17] found direct, solvent-dependent competition between CT state formation and SF. At the same time, some efficient SF dimers exhibit relatively little solvent dependence and appear to follow a direct $S_1 \rightarrow TT$ mechanism[10], raising the need for further systematic investigation of the nature and role of CT states in SF.

Here, we build on our previous study of 13,13'-bis(mesityl)-6,6'-dipentacenyl (DP-Mes, Fig. 1b), in which the two pentacenes are directly linked at the 6-position and are thus almost orthogonal[8,18]. We also investigate a similar molecule in which the mesityl side groups are replaced by triisopropyl-silylethynyl, 13,13'-bis((triisopropylsilyl)ethynyl)-6,6'-dipenta-cenyl (DP-TIPS, Fig. 1a). Through spectroscopic study of the solvent dependence of SF, we unambiguously demonstrate that CT states mediate triplet formation (Fig. 1). The SF rate and efficiency are directly tuned through solvent polarity, which modifies the energetic separation between CT and vertical locally excited (LE) states. Moreover, we find that the nature of the SF mechanism can be tuned by subtle manipulation of the molecular energy landscape through side-group engineering. In DP-Mes, the proximity of $S_1$ and CT states results in strong mixing between the two, yielding one-step SF through virtual CT states. In DP-TIPS, stabilisation of the singlet leads to a distinct two-step process with clear population of a CT intermediate, unique among efficient SF systems. Interestingly the population of this state can be suppressed by restricting the molecular geometry: in

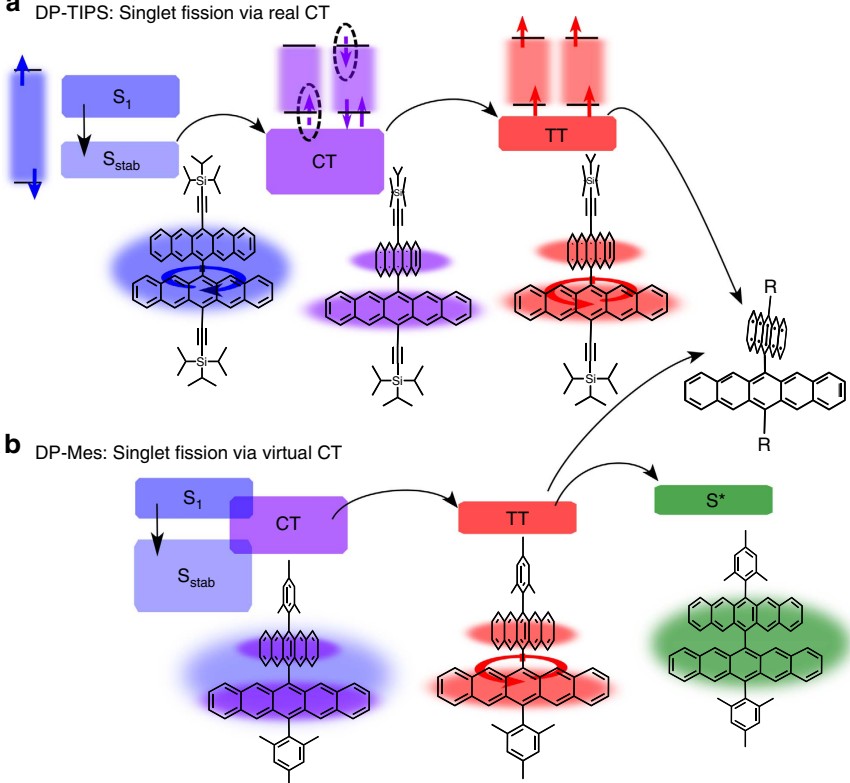

**Figure 1 | Tuneable SF pathways in pentacene dimers.** In DP-TIPS (**a**), the initial singlet state is rapidly stabilized into a relatively planar geometry $S_{stab}$. This process is quickly followed by solvent-polarity-dependent CT state formation, which favours a return to the orthogonal ground-state geometry. The CT state is intermediate to triplet pair (TT) formation. The kinetics of this final SF step depend on polarisation and reconfiguration of the solvent shell, and it is highly efficient in the orthogonal configuration. TT decays via non-radiative geminate recombination. In DP-Mes (**b**), the initial $S_1$ state is mixed with CT. Direct SF from this singlet state into TT competes with polarity-dependent solvent relaxation into $S_{stab}$. In non-polar media, subsequent TTA forms a planarized and emissive singlet state S*. In high-polarity media, no radiative TTA is seen and TT only decays by non-radiative geminate recombination. Coloured bars indicate the relative energetic positions of the states involved. Excited-state molecular geometries are inferred from spectroscopic data and comparison to literature.

a rigid polymer matrix, SF reverts to the standard one-step pathway. These findings resolve an important question about the mechanism of SF and suggest solubilizing groups as a powerful tool to fine-tune the SF process.

## Results

**Pentacene dimer absorption.** The absorption spectra of DP-Mes and DP-TIPS are presented in Fig. 2a,d. Both are characterized by 0-1/0-0 vibronic peak ratios ∼0.5, reduced from the corresponding Mes- and TIPS-substituted monomers P-Mes and P-TIPS (Supplementary Fig. 1). The spectra are also slightly broadened and shifted from the monomeric pentacenes: to the red in DP-Mes and to the blue in DP-TIPS. The overall shift is the result of competing effects between delocalisation into side groups, coupling via dark CT states and inter-pentacene dipole–dipole interactions.

We modeled the vertical $S_1$ excitations of both dimers and their corresponding monomers with time-dependent DFT (TDDFT) using the NWChem code[19]. Excitation energies and transition densities are summarized in Fig. 3a. We also performed constrained DFT calculations with ONETEP to model the pure electron transfer (ET) excitations[20–22]. Energies and electron–hole density plots are shown in Fig. 3b for vacuum; results with implicit solvent can be found in Supplementary Note 1 and Supplementary Fig. 2. As can be seen in Fig. 3a, the -TIPS side-groups participate more strongly in the $S_1$ excitation than the -Mes groups, a major contributor to the lower $S_1$ energies calculated and observed for the -TIPS versus -Mes molecules. The dimer $S_1$ states can essentially be interpreted as symmetric linear combinations of the corresponding monomer states, the transition dipoles of which are polarized along the short pentacene axis[23,24] (that is, along the connecting bond in the

dimers). In the dimer $S_1$ state these transition dipoles add constructively, producing a bright transition. In both dimers this is accompanied by a corresponding dark state, in which anti-parallel monomer dipoles cancel each other. This state is calculated to be 0.13 and 0.15 eV higher in energy for DP-Mes and DP-TIPS, respectively (Supplementary Table 1). This constitutes a significant exciton splitting with the interesting feature that the upper component is fully dark and therefore not detectable in absorption experiments. The calculations qualitatively reproduce the shifts observed from monomeric pentacene to the respective dimer—to the red for DP-Mes and to the blue for DP-TIPS—reflecting a competition between side-group and exciton-splitting effects. From monomer to dimer the ratio of side groups to pentacenes is reduced from 2:1 to 1:1. This reduction is expected to result in a blue-shift, particularly strong in DP-TIPS, which competes with the red-shift due to coupling of transition dipoles (similar in both molecules, approximately equal to exciton splitting). This results in an overall red-shift for DP-Mes and a small blue-shift for DP-TIPS. Altogether these theoretical results support the notion of significant dipole–dipole coupling between the individual pentacenes on dimerisation, in a way that is consistent with the observed shifts. This coupling energetically favours $S_1$ excitations which are fully delocalized over the dimers, in good agreement with the experimental data (see below). This should be the case even in the presence of some disorder, due to the robustness of the alignment of the dipoles associated with local excitations in each monomer.

Further evidence of the intramolecular coupling between pentacenes is the reduction of oscillator strength. It was previously observed in DP-Mes[8] that the peak molar extinction coefficient is half that of a monomeric pentacene, and the same is found for DP-TIPS (9700 l mol$^{-1}$ cm$^{-1}$ versus 21400 l mol$^{-1}$ cm$^{-1}$ in the monomer, in toluene). For comparison,

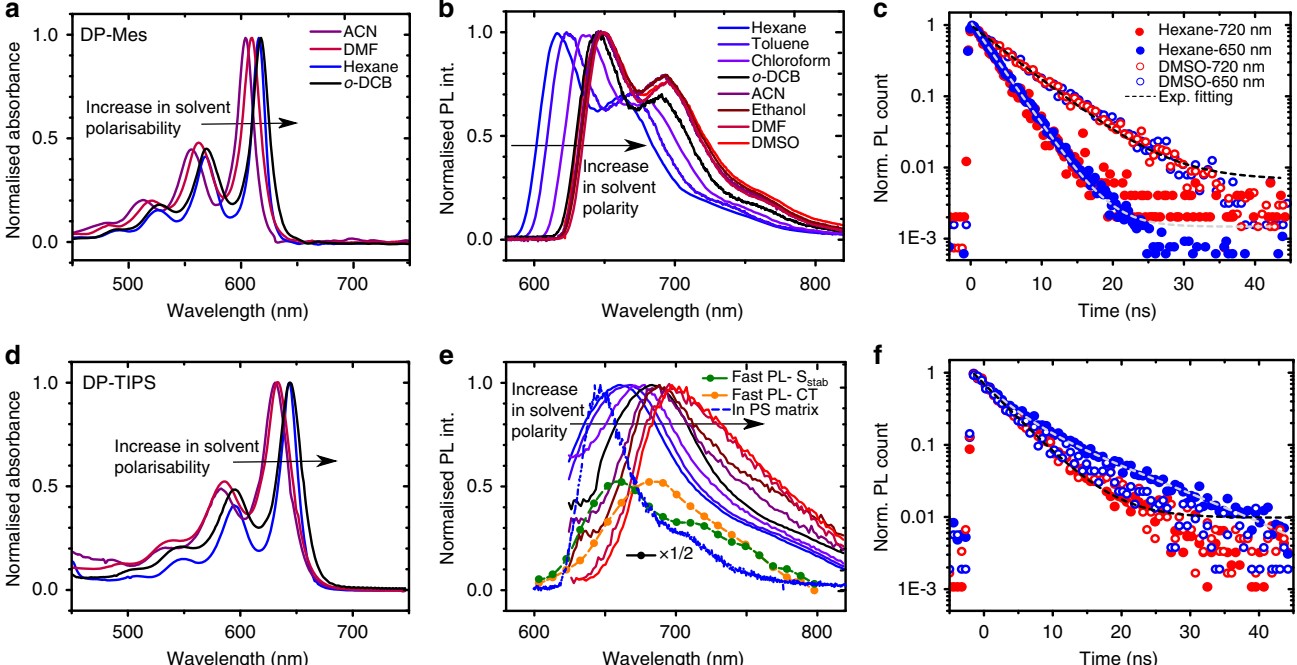

**Figure 2 | Solvent-dependent absorption and emission.** (**a**) DP-Mes and (**d**) DP-TIPS absorption in solvents spanning polarity and polarisability range. PL spectra of (**b**) DP-Mes and (**e**) DP-TIPS exhibit pronounced polarity dependence. Dashed line in (**e**) shows significant reduction in Stokes shift in rigid polystyrene matrix. Symbols are $S_{stab}$ and CT spectra extracted from a separate ultrafast PL experiment, in the first 3 ps following photoexcitation in o-DCB. (**c**) DP-Mes PL kinetics show uniform decay across the spectrum in non-polar and polar solvents. (**f**) DP-TIPS kinetics show spectrally-variant lifetimes in hexane (filled circles) indicating the presence of multiple emissive species. Uniform decay is observed in highly polar DMSO (open circles). Absorption spectra and PL kinetics for other solvents shown in Supplementary Figs 1 and 3–9. Legends are the same for top and bottom panels.

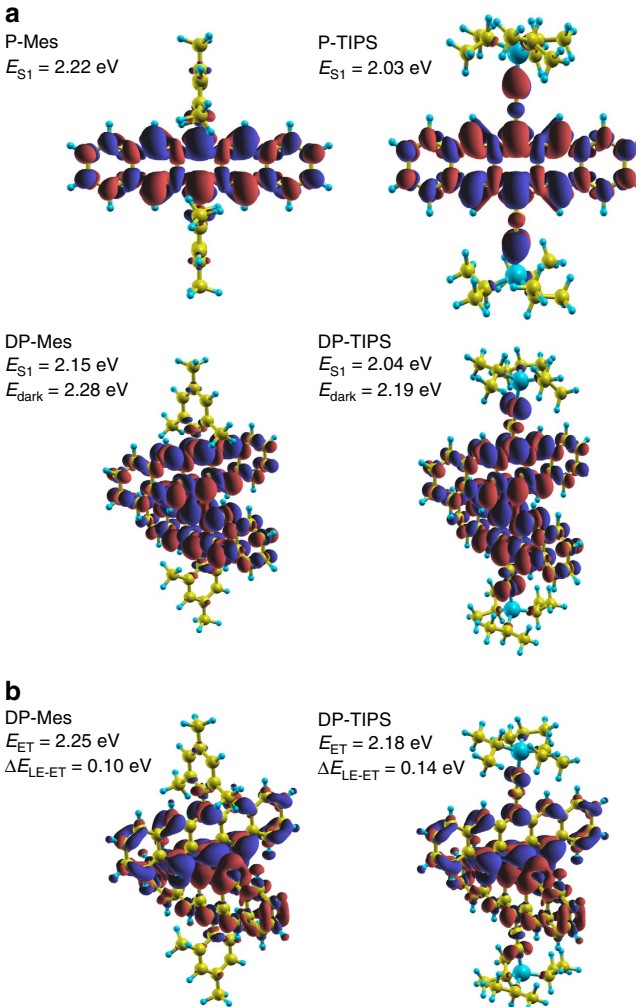

**a**

P-Mes
$E_{S1}$ = 2.22 eV

P-TIPS
$E_{S1}$ = 2.03 eV

DP-Mes
$E_{S1}$ = 2.15 eV
$E_{dark}$ = 2.28 eV

DP-TIPS
$E_{S1}$ = 2.04 eV
$E_{dark}$ = 2.19 eV

**b**

DP-Mes
$E_{ET}$ = 2.25 eV
$\Delta E_{LE-ET}$ = 0.10 eV

DP-TIPS
$E_{ET}$ = 2.18 eV
$\Delta E_{LE-ET}$ = 0.14 eV

**Figure 3 | Excited-state wavefunctions.** (**a**) $S_1$ excitation energies and transition densities for P-Mes, DP-Mes, P-TIPS and DP-TIPS, as determined from TDDFT[51]. The lower energy in -TIPS molecules can be largely attributed to delocalisation on the C-C triple bonds. The dimer $S_1$ state shown is the bright combination of monomer transitions. The energy of the dark combination with anti-parallel monomer transition dipoles is also indicated. (**b**) ET excitation energies and electron–hole densities for DP-Mes and DP-TIPS in vacuum, as determined from constrained DFT[51]. The effect of solvent is considered in Supplementary Fig. 2.

we note that most other molecular dimers investigated for SF exhibit weak electronic coupling and consequently the expected doubling of extinction coefficient relative to the monomer[9,11,25,26]. The observed behaviour suggests a significant degree of coupling in spite of a ground-state geometry which minimizes π–π inter-action between the two pentacenes[8,27–31]. We primarily attribute the interaction to J-type dipole-dipole coupling. In addition to the effects of side-group delocalisation, we propose that the absence of strong J-type spectral shifts is due to counteracting coupling mediated by dark CT states. It has been demonstrated that the combination of excitonic and CT coupling can result in 'null aggregates', exhibiting only slight red-shifts and minor distortions of the vibronic progression[32]. We infer similar behaviour from the solvent-dependent ultraviolet–Vis absorption. The spectra in Fig. 2 reveal no systematic dependence of the absorption edge ($\lambda_{1/2max}$ in Table 1) on solvent polarity, indicating a predominantly excitonic transition (Supplementary Fig. 1 for full solvent series; main parameters are listed in Table 1). As the polarity

is increased, lowering the energy of CT states (see below), the 0-0/0-1 vibronic peak ratio gradually increases, consistent with a reduction in the ground-state CT-mediated coupling[32]. This mechanism explains the substantial hypochromism (the loss of approximately 75% of the expected oscillator strength), as the $S_0$–$S_1$ transition is coupled to CT states with negligible oscillator strength. As the CT energy is reduced, coupling via this dark state weakens and we observe a systematic increase in extinction coefficient. The effect is small in DP-Mes, in which our time-resolved measurements demonstrate significant $S_1$-CT mixing at all polarities. In DP-TIPS the mixing is slightly weaker, resulting in higher and more tuneable extinction coefficient and distinct $S_1$ and CT states in time-resolved spectroscopy.

We note there is no spectral evidence of aggregation at the concentrations used, though aggregation can be induced in particular solvent mixtures (Supplementary Fig. 3). The time-resolved data below and extracted parameters in Table 1 reveal gradual changes of dimer photophysics with solvent polarity and polarisability, rather than abrupt changes due to the onset of aggregation. We thus consider all solutions in this study to consist of well-isolated molecules, with all measurements pertaining to intramolecular properties.

**Pentacene dimer photoluminescence**. We detect pronounced differences between the two molecules in steady-state photo-luminescence (PL), which is strongly sensitive to solvent polarity (Fig. 2b,e). The DP-Mes emission spectrum red-shifts without any appreciable change in shape, maintaining the same clear vibronic structure. Time-resolved PL kinetics (Fig. 2c, Supplementary Fig. 4) reveal uniform decay across the spectrum, confirming the presence of only a single long-lived emissive species in all solvents. The PL lifetime varies from 2 to 3 ns in hexane to ∼5 ns in polar media, and it has been previously demonstrated that these lifetimes correspond to two distinct states[8]. Transient absorption (see below) confirms that the emissive species in non-polar solvent is a planarized singlet state S* formed by triplet–triplet annihilation (TTA), while in polar media it is a singlet exciton unable to undergo SF due to solvent stabilisation ($S_{stab}$). This behaviour is reflected in the solvent-dependent PL quantum yield (Table 1). In toluene and hexane the annihilation process to form S* is efficient and results in substantial delayed emission. At higher polarity, S* is inaccessible but solvent stabilisation becomes increasingly competitive with SF, enhancing the population of long-lived emissive $S_{stab}$.

Surprisingly, the PL spectra of DP-TIPS (Fig. 2e) exhibit no strong vibronic structure, and they change shape as well as red-shift with increasing polarity. The underlying dynamics (Fig. 2f) are distinctly more complex than in DP-Mes. In low-polarity solvents (filled circles) we detect two lifetimes: 10.8 ns at the blue spectral edge and a faster 4.8 ns in the red, demonstrating the presence of two independent emissive species. As polarity increases, the difference between them becomes less discernible, eventually giving the uniform single-exponential decay seen in DMF and DMSO (Fig. 2f, Supplementary Fig. 5). To disentangle the two components, we performed spectral decomposition on detailed PL decay maps (Supplementary Figs 6 and 7). The first consists of vibronic peaks at 650 and 690 nm, resembling the initial spectrum detected using ultra-fast transient grating PL (Fig. 2e, Supplementary Fig. 8)[33]. This component has a lifetime of 11 ns and shows no solvent dependence. By analogy to DP-Mes and the TA results discussed below, it is assigned to singlet excitons which are unable to undergo SF ($S_{stab}$). The second species is broad, without prominent vibronic features, and decays with a shorter, solvent-independent 5 ns lifetime. The spectral position of this species varies strongly with polarity: it overlaps

**Table 1 | Solvent properties and solvent-dependent absorption, PLQE and triplet yield parameters of DP-Mes and DP-TIPS.**

| Solvent | Polarity* | Polarisability $(\varepsilon_0)$† | Viscosity (cP)‡ | Absorbance DP-Mes | | | PLQE DP-Mes (%) | Triplet yield DP-Mes (% ± 15%) | Absorbance DP-TIPS | | | PLQE DP-TIPS (%) | Triplet yield DP-TIPS (% ± 15%) |
|---|---|---|---|---|---|---|---|---|---|---|---|---|---|
| | | | | $\lambda_{1/2\ max}$ (nm) | 0-0/0-1 | Mol. ext. coeff. (l mol$^{-1}$ cm$^{-1}$) | | | $\lambda_{1/2\ max}$ (nm) | 0-0/0-1 | Mol. ext. coeff. (l mol$^{-1}$ cm$^{-1}$) | | |
| Hexane | 0.54 | 11.9 | 0.31 | 624 | 0.43 | 7500 | 7.2 | 197 | 654 | 0.41 | 8800 | 7.6 | 177 |
| Toluene | 0.66 | 12.4 | 0.56 | 625 | 0.45 | 7900 | 4.2 | 193 | 656 | 0.43 | 9700 | 3.4 | 179 |
| Chloroform | 0.79 | 8.5 | 0.54 | 621 | 0.46 | 8000 | 5.6 | 179 | 652 | 0.47 | 10800 | 1.5 | 191 |
| o-dichlorobenzene (o-DCB) | 0.82 | 13.2 | 1.32 | 628 | 0.46 | 8150 | 8.3 | 164 | 659 | 0.48 | 11300 | 0.2 | 193 |
| Ethanol | 0.85 | 5.1 | 1.07 | 615 | 0.48 | 8200 | 11.1 | 152 | 647 | 0.49 | 12200 | 2.5 | 182 |
| Acetonitrile (ACN) | 0.90 | 4.9 | 0.34 | 612 | 0.46 | 8220 | 13.9 | 146 | 645 | 0.49 | 12700 | 2.3 | 175 |
| N,N-dimethylformamide (DMF) | 0.95 | 7.9 | 0.79 | 619 | 0.50 | 8300 | 17.6 | 131 | 650 | 0.53 | 13200 | 4.8 | 165 |
| Dimethyl sulfoxide (DMSO) | 1.00 | 8.0 | 1.99 | 621 | 0.50 | 8400 | 20.3 | 122 | 651 | 0.55 | 14500 | 5.6 | 163 |

*Adopted from ref. 52.
†Taken from ref. 53.
‡Adopted from ref. 54.

with $S_{stab}$ in hexane, shifts to 685 nm in o-DCB and further to 710 nm in DMF. Such broad, featureless and polarity-sensitive emission are typical signatures of CT states[34], and the shifts follow the behaviour predicted by cDFT calculations (Supplementary Fig. 2). We thus assign this feature to an emissive intramolecular CT state, resembling a radical anion localized on one pentacene and a radical cation on the other (see below), though there must be sufficient wavefunction overlap between these to enable weak emission. This state is intermediate to efficient SF in DP-TIPS.

Our results demonstrate that the relative population of emissive species is highly polarity-dependent in both dimers. Solvent polarity influences their excited-state energetic landscapes and accordingly SF. At the same time, we see that the solubilizing groups unequivocally influence dimer luminescence, from distinct vibronic emission in DP-Mes to complex, multi-species emission in DP-TIPS. To build a more detailed picture of how these factors modifypentacene dimer photophysics, we apply transient absorption (TA) spectroscopy.

**Solvent dependence on SF in DP-Mes.** In Fig. 4 we present typical TA data for DP-Mes at the extremes of the polarity series, hexane (a) and DMSO (b). As we have previously shown[8], the TA spectra of DP-Mes can be decomposed into three distinct species: singlet excitons $S_1$, triplet excitons TT and the emissive state formed via TTA, $S^*$ (Fig. 4c). Initial $S_1$ can be unambiguously identified by the stimulated emission band (SE, $\Delta T/T > 0$) at ~650 nm, which matches prompt photoluminescence detected with an ultrafast transient grating technique[8,33,35] (Supplementary Fig. 9). The spectrum of triplet pairs contains precisely the same sharp photo-induced absorption peaks (PIA, $\Delta T/T < 0$) at 650, 815 and 915 nm observed in a separate triplet sensitisation experiment[8]. The final state, assigned to $S^*$, has a unique PIA signature as well as slightly red-shifted SE, confirming its identity as a lower-energy emissive singlet.

Using these species-associated spectra, the raw TA kinetics (symbols, Fig. 4d,e) can be converted into population dynamics (filled regions). In all solvents (full series presented in Supplementary Figs 10 and 11) these reveal direct formation of triplets from singlet decay, through fast intramolecular SF (<2 ps), and from the magnitude of TT features we directly extract the solvent-dependent triplet yield shown in Table 1. It is evident from the population kinetics in Fig. 4 that the triplet

formation rate varies with solvent, as can be directly tracked through the singlet SE (Fig. 4f). The $S_1$ lifetime is systematically shortened with increasing polarity, from 1.17 ps in hexane to ~300 fs in the most polar solvents. As previously noted[8,12], this direct correlation of SF time constant to solvent polarity suggests triplet formation is mediated by CT states, though these are not directly resolved. However, we highlight the intriguing spectral shape of $S_1$ in the NIR. Such well-defined structure is unusual for pentacene singlet spectra[8,36] (Fig. 4c), but we find striking correlation with the expected PIA spectrum of a CT state determined from chemical oxidation and reduction (dashed line, Fig. 4c; anion and cation spectra in Supplementary Fig. 12). We thus propose that the initial singlet in DP-Mes is strongly mixed with a nearby CT state. We return to the implications of this below.

The kinetics in Fig. 4 also illustrate the balance of emissive species identified in time-resolved PL. As triplets excitons decay in low-polarity solvent, we observe new PIA bands and SE red-shifted from that of the initial singlet, features assigned to $S^*$. From the broadened and red-shifted PL of $S^*$ we infer it is relaxed from the original orthogonal conformation. We previously observed that $S^*$ is suppressed in a rigid polystyrene matrix[8]. To further probe the role of geometric relaxation we have studied SF and subsequent $S^*$ formation as a function of viscosity (Supplementary Fig. 13). Increasing the concentration of polystyrene in toluene solutions does not affect the SF kinetics and only slightly increases the triplet lifetime. The most noteworthy change is the reduction and eventual suppression of $S^*$ formation at high viscosity, detected through TA and PL. These experiments confirm that the radiative TTA pathway requires a degree of large-scale intramolecular motion, presumably into a lower-energy planar geometry.

In higher-polarity solvents, the progressive enhancement of SF rate is coupled with a reduction in triplet yield (Table 1 and Supplementary Fig. 14). This unusual behaviour arises from a competing decay channel for singlet excitons, solvent stabilisation. The effect can be visualized in the final time-slice of DP-Mes in DMSO (Fig. 4b, red), by which point the triplet population has almost fully decayed. The remaining PIA closely resembles the initial singlet spectrum and is accompanied by red-shifted SE, that is, it represents the same electronic state that was initially photoexcited, but relaxed to a lower energy. These are signatures of $S_{stab}$ and manifest as the plateau in the singlet dynamics (Fig. 4e), showing that up to 50% of the population is unable to

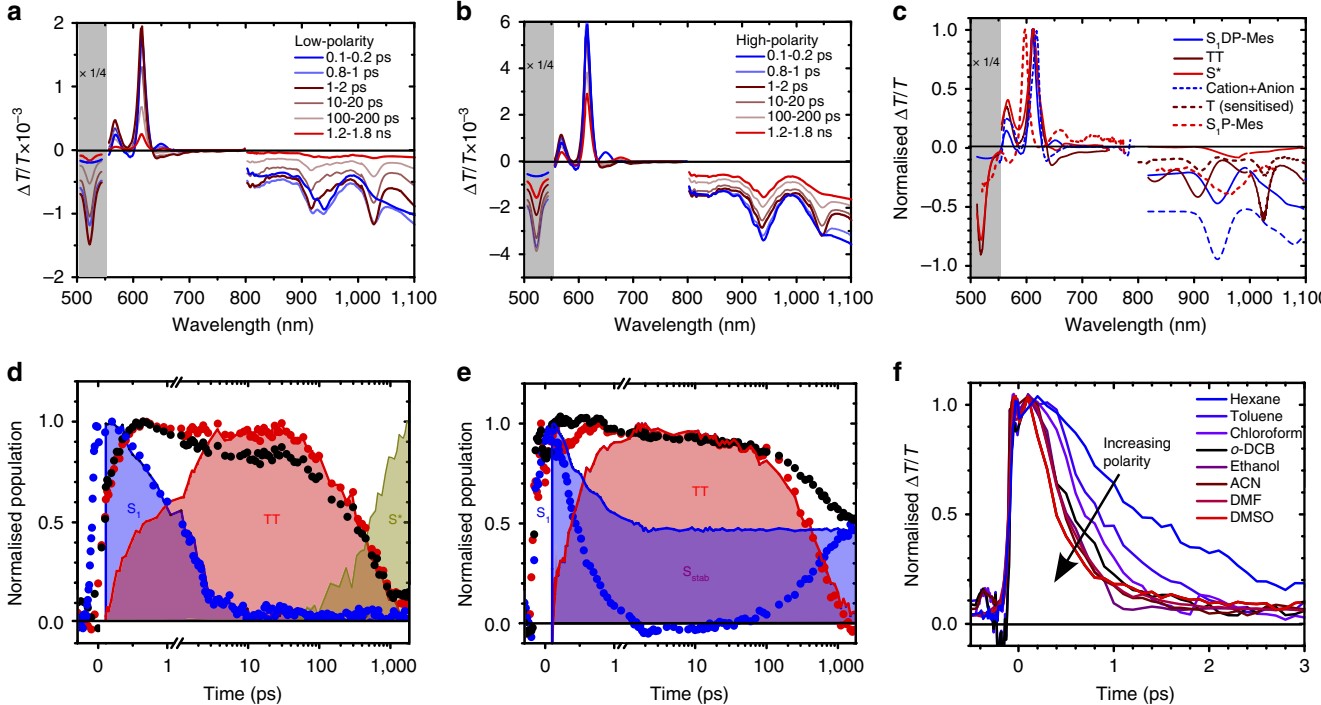

**Figure 4 | SF in DP-Mes.** TA spectra of DP-Mes in (**a**) low-polarity hexane and (**b**) high-polarity DMSO at indicated time delays. The strong triplet PIA at $\sim 550$ nm is scaled by a factor of $\frac{1}{4}$ for clarity. (**c**) Species extracted from spectral decomposition all exhibit distinct spectral features in the NIR spectral range. The PIA of TT gives an excellent match to the triplet spectrum generated through sensitisation[8]. Both $S_1$ and $S^*$ exhibit stimulated emission and unique PIA bands. The cation + anion spectrum was generated from separate chemical oxidation and reduction experiments and closely matches the $S_1$ PIA in the NIR. Species are similar across all solvents used in the study, only the result from toluene is presented. TA decay kinetics (symbols) in (**d**) hexane and (**e**) DMSO were averaged over 520–530 nm (triplet PIA, red), 610–620 nm (ground-state bleach, black) and 665–670 nm ($S_1$ SE, blue). Normalized species population kinetics (filled regions) show strong agreement with raw kinetics. SF in hexane is quantitative, and later triplet decay matches the rise of emissive $S^*$. In DMSO, 40–50% of the $S_1$ population fails to form triplets and is trapped in $S_{stab}$. No signatures of $S^*$ are detected. (**f**) Early-time dynamics of $S_1$ SE decay at 665–670 nm show the strong dependence of $S_1$ lifetime and SF rate on solvent polarity. Full TA data for all solvents can be found in Supplementary Figs 10 and 11.

undergo SF in the most polar solvent. Both the long-lived population and the degree of stabilisation of $S_{stab}$ vary directly with solvent polarity, with its energy spanning a range of almost 100 meV (Fig. 2b). This emissive state can be directly generated from sub-band gap excitation, resulting in long-lived singlets instead of SF (Supplementary Fig. 15). At very low excitation energy (670 nm in hexane), a small population of $S^*$ can also be directly excited, a consequence of room-temperature structural dynamics which yield a small population of relatively planarized dimers.

**SF in DP-TIPS**. Analogous measurements were performed on DP-TIPS in the same solvents (Fig. 5). Following resonant photo-excitation in hexane, we observe positive $\Delta T/T$ signals at 635 and 680 nm which match the ground-state absorption and PL spectra and can be assigned to ground-state bleach (GSB) and SE, respectively. The SE signature of $S_1$ is accompanied by PIA at 850 and 1325 nm. These features decay concurrently with a lifetime of 1.23 ps. On long timescales ($>10$ ps) a trio of sharp absorption peaks at $<550$ nm, 880 nm and 1010 nm appear and persist for hundreds of ps. These longer-lived features precisely match the triplet PIA spectrum generated through sensitisation[37] (Fig. 5g and Supplementary Fig. 16). Such fast and high-yield triplet formation ($177\% \pm 15\%$ in hexane, see Supplementary Note 2 for details) demonstrates that DP-TIPS also undergoes SF. A separate proof is the rapid timescale for triplet decay[8–10,37,38]. A single DP-TIPS triplet generated though sensitisation has lifetime

$>1$ μs, whereas the triplets formed by direct excitation typically exhibit sub-ns lifetimes insensitive to polarity, polarisability or concentration (Fig. 5i, Supplementary Figs 17 and 18). In our dilute solutions, this behaviour is only possible through intramolecular TTA. This, in turn, requires intramolecular SF as in DP-Mes, resulting in one triplet exciton localized on each pentacene in the dimer.

Close examination of the intermediate timescales reveals a crucial difference from DP-Mes. Between the decay of $S_1$ and formation of TT, spectral features appear which cannot be assigned to either $S_1$ or TT. Decay of $S_1$ yields sharp PIA bands on either side of the main GSB peak, resembling an electroabsorption Stark effect. This correlates with new red-shifted SE $>700$ nm and shifted NIR PIA peaks. Spectral decomposition confirms that the state with these features is intermediate to triplet formation (filled regions, Fig. 5d), demonstrating that SF in DP-TIPS entails a three-state progression as illustrated in Fig. 1. A separate ultrafast transient grating PL measurement identified the distinct PL signatures of $S_1$ and the broad, delayed CT state emission in the first 10 ps following photoexcitation (Fig. 2e and Supplementary Fig. 8). The kinetics of these species (open symbols, Fig. 5d–f) provide an excellent match to the population kinetics extracted from TA, enabling assignment of the intermediate as the emissive intramolecular CT state. Our assignment is further confirmed through separate chemical oxidation and reduction experiments, which reveal a good match with a radical anion-radical cation pair across the full spectral range (Fig. 5h). This is a direct and unambiguous observation of a populated CT intermediate in an

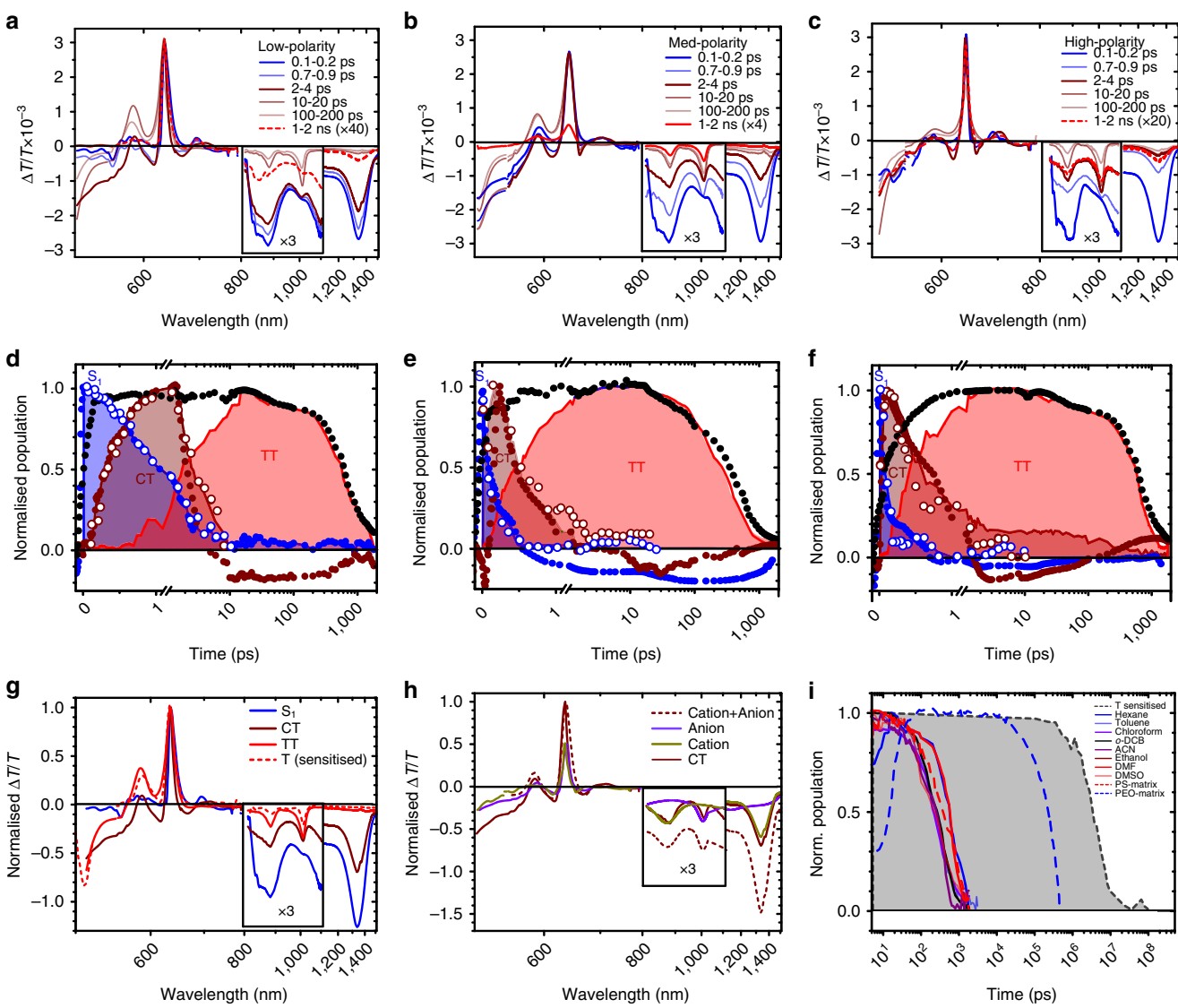

**Figure 5 | SF in DP-TIPS.** TA spectra of DP-TIPS in (**a**) low-polarity hexane, (**b**) medium-polarity *o*-DCB and (**c**) high-polarity DMF at indicated time delays. The dashed line showing the spectrum measured on long time-scales and the entire NIR spectral region have been magnified for clarity. Corresponding TA decay kinetics (filled symbols) in (**d**) hexane, (**e**) *o*-DCB and (**f**) DMF reveal the presence of multiple species. Kinetics are averaged over ranges 615–620 nm (CT PIA, brown), 635–645 nm (GSB, black) and 680–690 nm ($S_1$ SE, blue). $S_1$ and CT kinetics are closely matched by results from the separate transient grating PL experiment (open symbols, $S_1$: blue and CT: brown). Population kinetics extracted from spectral decomposition (filled regions) illustrate a sequential $S_1 \rightarrow CT \rightarrow TT$ progression, with residual $S_{stab}$ population in hexane and residual CT population in DMF. (**g**) Three distinct spectral species extracted from TA data (solid). The peak positions of the triplet pair TT match those determined from a separate sensitisation experiment (dashed), though the ratio of PIA to GSB is reduced in sensitisation due to the presence of only one triplet. (**h**) The spectral shape of the extracted CT PIA can be well reproduced as a combination (dashed) of anion and cation spectra from chemical oxidation and reduction. Note it is not physically possible to reproduce the SE band at 700 nm without directly generating CT states. (**i**) Comparison of normalized TT population kinetics for the full range of solvents and polymer matrices studied here to the intrinsic single-triplet lifetime from sensitisation (filled). The decay of TT is orders of magnitude faster due to rapid intramolecular TTA, except in the unique PEO matrix. Full TA, transient grating PL and sensitisation results are presented in Supplementary Figs 8, 16–19, 21 and 22.

efficient SF system. This is not a pure, diabatic electron-transfer state, but rather a partial symmetry-breaking charge separation between the dimer, as has been suggested in 9,9-bianthryl[28] and other systems[17,39]. This endows it with non-zero oscillator strength, allowing both photoluminescence and, analogous to CT states in donor-acceptor systems[40,41], sub-gap excitation (Supplementary Fig. 19). It is interesting to note the striking similarity in spectral shape (if not position) between this CT state and the initial $S_1$ signature in DP-Mes (Supplementary Fig. 20). We return to this issue below.

**Solvent dependence on SF in DP-TIPS.** The same progression of states is observed in all solvents; we have highlighted in Fig. 5 the results from hexane (a,d), *o*-DCB (b,e) and DMF (c,f) which illustrate the full range of DP-TIPS photophysics (remaining solvents in Supplementary Figs 21 and 22). Just as in DP-Mes, increased polarity results in faster decay of $S_1$, though in DP-TIPS this decay forms CT rather than triplet pairs, and it becomes more efficient as the rate is enhanced. In hexane, the final state observed is not triplets but a small population of residual $S_1$ (dashed lines, Fig. 5a). These states were unable to form CT or,

subsequently, TT, and decay with a lifetime of 9.1 ns, in good agreement with the long-lived $S_{stab}$ emission observed in non-polar solvents (Fig. 2f and Supplementary Fig. 5). As solvent

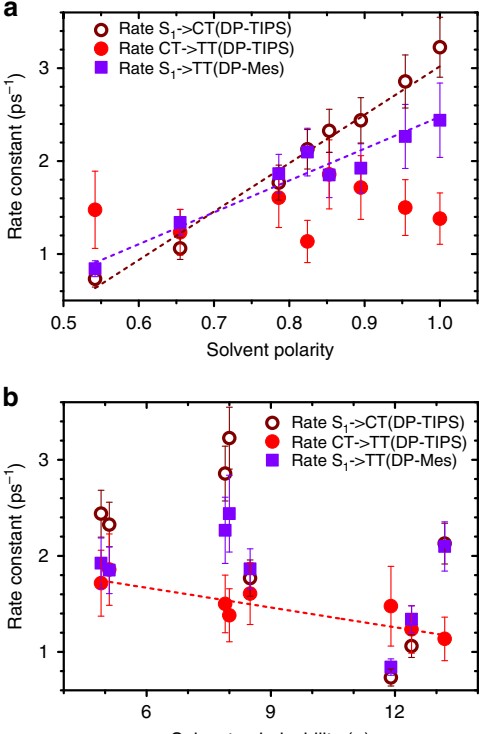

**Figure 6 | Solvent-dependent SF rate constants.** (**a**) The SF rate for DP-Mes ($S_1 \rightarrow TT$, purple) and the rate of $S_1 \rightarrow CT$ conversion in DP-TIPS (brown) both depend linearly on solvent polarity. The polarity dependence of $CT \rightarrow TT$ conversion (red) in DP-TIPS is complex, (**b**) but the rate exhibits moderate correlation with polarisability. Lines are linear fits. Rates were determined from population kinetics and yields extracted from TA, as described in Methods and Supplementary Note. Error bars reflect propagation error from calculating triplet yield and exponential fitting of the conversion rate.

polarity is increased, the amount of $S_{stab}$ detected in TA steadily decreases, and already in $o$-DCB the $S_1 \rightarrow CT$ process appears to be quantitative.

The second process, conversion of the CT intermediate into triplet pairs, also shows strong solvent dependence. At the highest polarities (DMF and DMSO), a significant fraction of CT population can be detected at long time delays after most of the triplets have decayed (dashed line, Fig. 5c). These longer-lived CT states form on early timescales and never undergo SF, and their 4.5 ns decay lifetime matches the CT emission identified in Fig. 2. As the polarity is decreased from these extreme solvents, the amount of residual CT is reduced until $o$-DCB, in which the $CT \rightarrow TT$ process is essentially quantitative. The solvent dependence of the rate is not straightforward; however (Fig. 6a, Supplementary Table 2), with no clear relation to polarity. Interestingly, the rate constant displays moderate correlation with solvent polarisability (Fig. 6b). The process of TT formation can be suppressed through sub-gap excitation into low-lying CT states available in some solvents. Scanning the excitation wavelength for DP-TIPS through the polarity series (Fig. 7 and Supplementary Fig. 23), we observe that beyond the threshold of 660 nm we increasingly directly excite CT rather than $S_1$. The triplet yield stays constant up to 680 nm ($\Delta E_{CT-TT} \sim -50$ meV) and decreases sharply for longer wavelengths due to the stabilisation of CT in higher polarity solvent.

The solvent-dependent behaviour of $S_1 \rightarrow CT$ and $CT \rightarrow TT$ dynamics combines to explain the triplet yield and PL quantum efficiency data in Table 1. At low polarity, the triplet yield is reduced and the PLQE enhanced because a moderate population of singlet states is unable to form the CT intermediate prior to solvent stabilisation, leading to $S_{stab}$ emission. At high polarity, the singlets are rapidly quenched but a substantial population remains trapped in the weakly emissive CT state on long timescales, giving a comparably high PLQE and lower triplet yield. Only in the intermediate cases are both loss processes inefficient, enabling high triplet yields and low PLQE in chloroform and $o$-DCB.

**Effect of matrix rigidity on SF in DP-TIPS.** To study the role of conformational change on the SF kinetics in DP-TIPS, we embedded it at low concentration in polymer matrices with

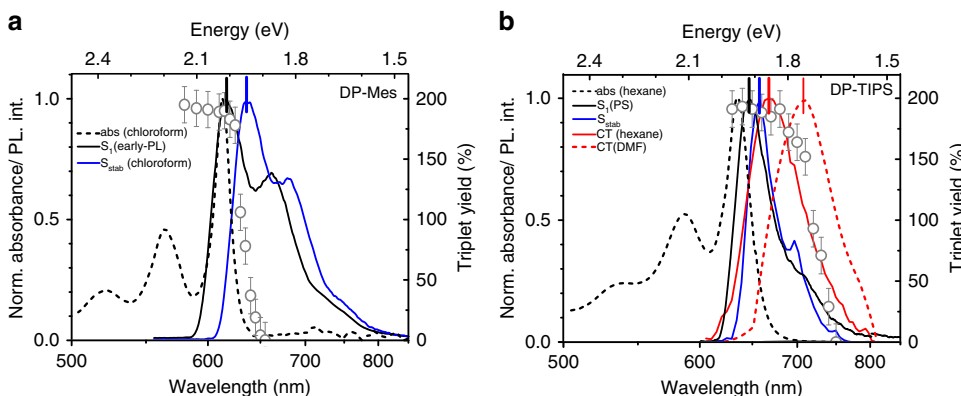

**Figure 7 | Excitation-dependent triplet yield.** (**a**) Absorption (dashed) and prompt (black) PL spectra of DP-Mes in hexane and the corresponding extracted triplet yield (circles) as function of excitation energy. Full TA datasets presented in Supplementary Fig. 15. Delayed PL spectrum in chloroform (blue) presented for reference. (**b**) Absorption of DP-TIPS in hexane (dashed black), steady-state PL spectra in polystyrene matrix (unrelaxed $S_1$, black) and DMF (CT, dashed red) and extracted PL components in hexane (prompt = relaxed $S_1$, blue; delayed = CT, red). The corresponding triplet yield in a series of solvents (circles) follows the CT band rather than ($S_1$) absorption edge, revealing SF from direct CT excitation to a threshold of 750 nm. Triplet yields are approximated by comparing the CT to TT excited-state absorption strength, benchmarked against the ratio in $o$-DCB following resonant excitation, where nearly quantitative SF is seen. Full TA datasets presented in Supplementary Fig. 23. Vertical bars at the peaks of PL spectra indicate the assigned energy of the emissive state. Errors bars reflect the error in calculating triplet yield.

different glass transition temperatures, giving a room-temperature rigidity series PS > poly(vinylacetate) (PVAc) > polyethylene-oxide (PEO). Surprisingly, in the rigid PS matrix SF proceeds directly from $S_1$ to TT, with no evidence in TA (Fig. 8a,b) or PL (Fig. 2e) of the CT state. In the intermediate matrix, we see a mixture of this direct process with the CT-mediated pathway. On longer timescales we observe some CT→TT conversion, but a large proportion of CT states are trapped, limiting the triplet yield to ~150%. This effect is exacerbated in PEO matrix, in which only a small fraction of CT states are able to form triplets, resulting in a significant increase in the CT emission (Supplementary Fig. 24) and great reduction in triplet yield (~60%). Though the yield is low, the triplet lifetime is greatly enhanced in this matrix compared with solutions or rigid films (900 ns versus < 1 ns). The significant impact of matrix rigidity highlights the importance of molecular conformation in determining both the pathway of SF and the decay of triplet pairs.

## Discussion

Figure 9 summarizes the SF process in DP-Mes and DP-TIPS for the purpose of discussion. We consider that the photophysics of both DP-TIPS and DP-Mes can be well described by the established model of CT-mediated SF[13–15] and the mixing of CT configurations into $S_1$. In DP-TIPS, we observe this CT state as a distinct intermediate in efficient SF. We are able to tune the fission process through solvent polarity, which directly modulates the CT energy. In non-polar media like hexane, the CT state is higher in energy than $S_{stab}$ and relatively slow $S_1 \rightarrow$ CT conversion competes with solvent relaxation. As the polarity is increased,

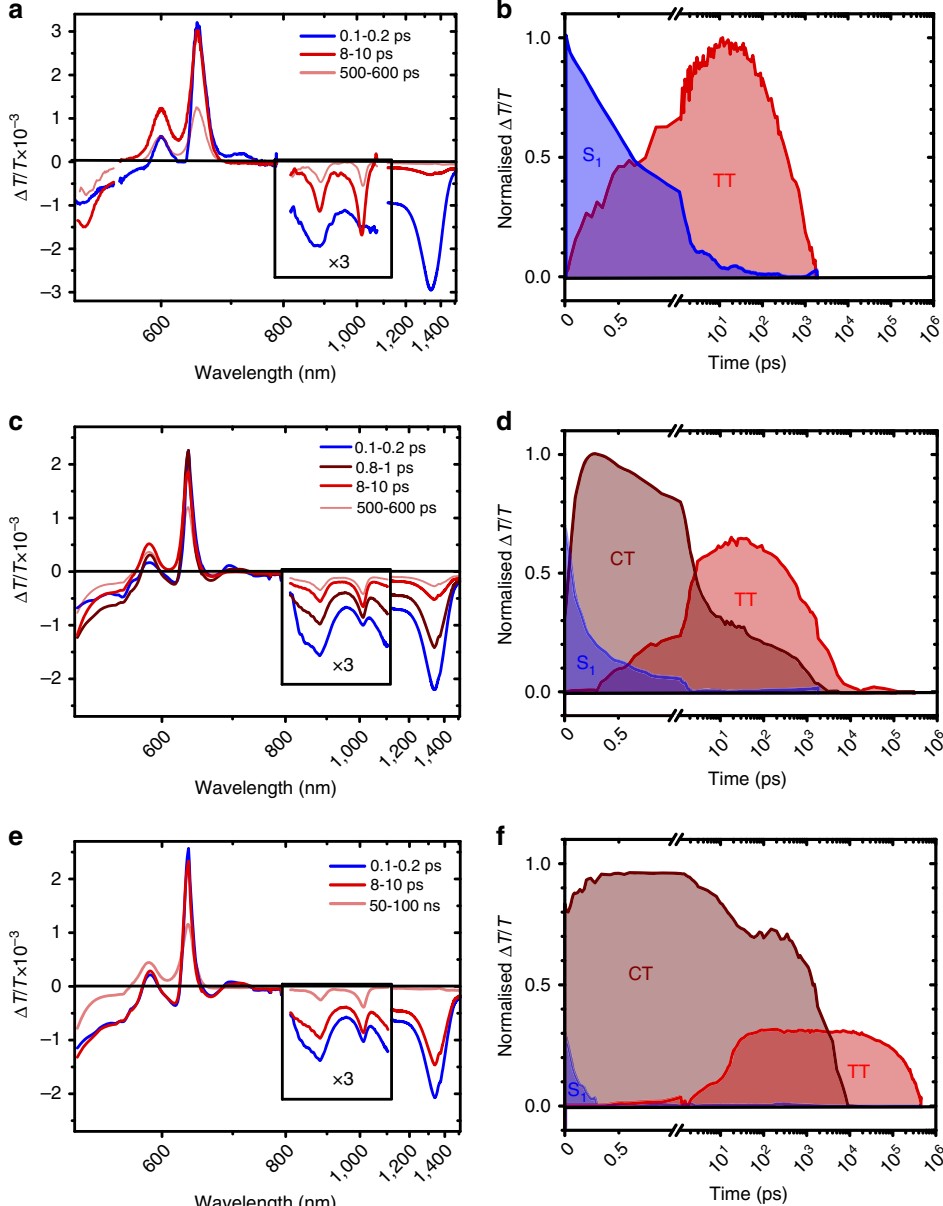

**Figure 8 | SF in DP-TIPS in polymer matrix.** TA spectra of DP-TIPS embedded in polymer matrices with different glass transition temperatures: (**a**) PS (**c**) PVAc and (**e**) PEO. The same features of $S_1$ and TT are evident in the matrices as in solution. The NIR spectral range has been scaled by a factor of 3 for clarity. The characteristic dual PIA of CT at 620 and 650 nm can only be detected in PVAc and PEO matrices. The population kinetics for $S_1$, CT and TT obtained from spectral decomposition are presented in (**b,d,f**). The CT population is significantly stabilized in both PVAc and PEO, and the triplet lifetime is also greatly enhanced in the relatively soft PEO matrix.

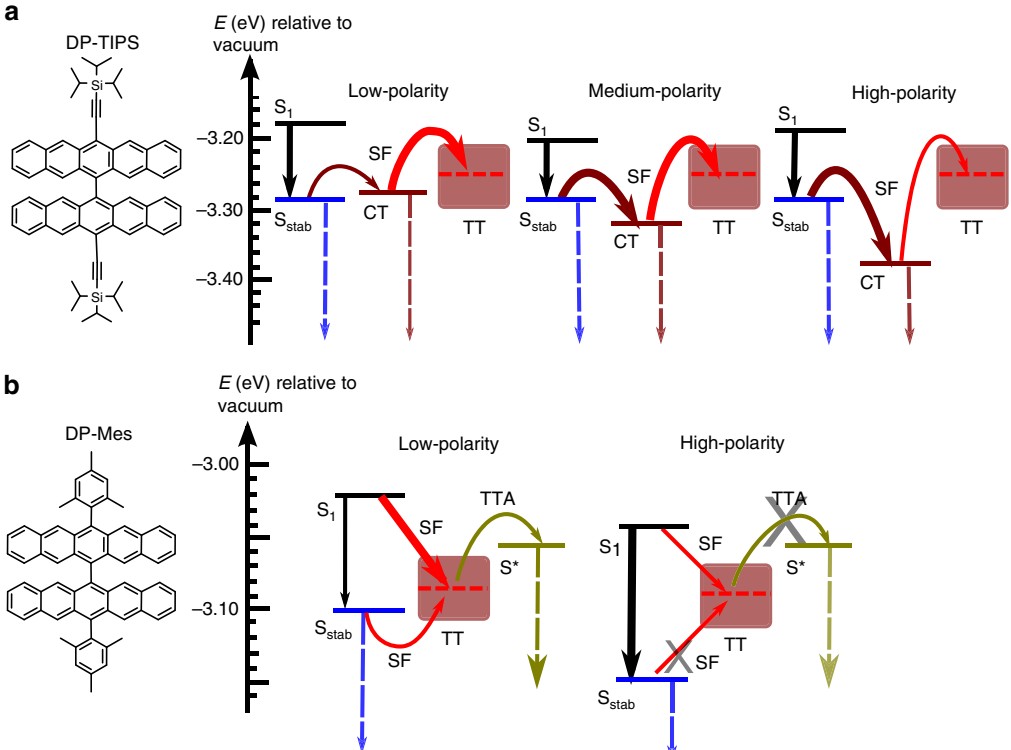

**Figure 9 | Summary of solvent-dependent SF in orthogonal pentacene dimers.** (**a**) In DP-TIPS, rapid initial solvent stabilisation of $S_1$ is followed by CT state formation. The CT state is intermediate to TT formation, and its energy can be tuned with solvent polarity. At very high polarities CT functions as an energetic trap, severely reducing the triplet yield. No radiative TTA is observed in DP-TIPS. (**b**) In DP-Mes, direct SF competes with solvent relaxation into $S_{stab}$. In non-polar media, solvent stabilisation is weak and all singlet states are able to undergo SF, while subsequent TTA forms a planarized and emissive singlet state S*. In high-polarity media, solvent stabilisation results in long-lived and lower-energy $S_{stab}$ states, and no radiative TTA occurs. Dashed arrows indicate radiative transitions, and solid arrows denote efficient (thick) and inefficient (thin) excited-state conversions. Energies are taken from absorption ($S_1$), fluorescence ($S_{stab}$, CT and S*) and phosphorescence (Supplementary Fig. 27, TT $\sim 2 \times T_1$).

CT drops below $S_{stab}$ (itself invariant with polarity) and this first step of SF becomes more rapid and efficient. The second step, conversion of CT→TT, also initially becomes more rapid as the CT energy is reduced, presumably due to improved matrix elements for the conversion[13]. However, at the highest polarities this state can actually relax below the energy needed to generate TT, thus the rate and efficiency decrease. In DP-Mes the same basic principles apply, but the role of CT reverts to a virtual intermediate. Again, as the CT state lowers in energy through increased polarity, the matrix elements governing $S_1$→TT conversion (and thus the rates) are increased. At moderate to high polarity, an increasingly large fraction of the population is energetically stabilized against SF, due to a reduction in the $S_{stab}$ energy.

This highlights the peculiar observation that solvent polarity directly modulates the emissive CT energy (broad PL spectra) in DP-TIPS, but modifies the relaxed singlet energy (vibronic-structured PL spectra) in DP-Mes. We propose that the discrepancy between the two dimers can be explained through the admixture of CT into $S_1$. We recall that the sharply structured TA spectrum of $S_1$ in DP-Mes more closely resembles the cation + anion combination spectrum, and the CT state of DP-TIPS, than it does the broad singlet signatures of DP-TIPS or either monomer[8,36]. A simple estimation of the energy of the pure electron-transfer (ET) state from the ionisation potential and electron affinity gives a value of 2.08 eV for both dimers[8,18], which is closer to the measured $S_1$ of DP-Mes (2.03 eV) than DP-TIPS (1.92 eV). Likewise, TDDFT and c-DFT calculations of both dimers in the ground-state geometry indicate a smaller pure

ET to locally excited (LE) energy gap for DP-Mes than DP-TIPS (Fig. 3). It should be the case in any solvent that the relevant charged state is closer to the singlet energy of DP-Mes, though we stress that the diabatic LE and ET states in the calculations do not directly correspond to the adiabatic $S_1$ and CT states observed in our spectroscopic measurements. We conclude that the initial state in DP-Mes is a singlet strongly mixed with the nearby ET states, nevertheless retaining its predominantly excitonic character. This ET contribution to the state results in wide variation of the $S_{stab}$ energy with solvent polarity and enables rapid SF in a one-step process by mediating the coupling to the triplet manifold[10]. Indeed, this picture is similar to that for pentacene thin films, in which the CT component considered necessary to drive SF is present in the initial singlet state[15]. The TIPS groups in DP-TIPS, on the other hand, significantly lower the singlet energy with little effect on the ET state. As a result, the initial state is more purely Frenkel excitonic, and only subsequent geometric distortion enables transfer into the distinct CT state.

Importantly, our results show that the CT-mediated SF pathway is closely linked to molecular geometry in both dimers. The rigid PS matrix greatly restricts molecular motion and holds the dimer in its orthogonal ground-state geometry. In DP-TIPS, these conditions allow fast and efficient SF without any losses in the form of $S_{stab}$ or CT states, and SF appears to be direct from $S_1$. Polymer matrix[8] and high-viscosity solution (Supplementary Fig. 13) measurements of DP-Mes show comparable effects, pointing to the orthogonal geometry as the most favourable for SF.

We propose that this geometry dependence also helps to explain the significant observed difference in SF dynamics. In the analogous anthracene dimer 9,9′-bianthryl the equilibrium geometry for the Frenkel-excitonic singlet state has a ~70° dihedral angle between acenes[28,42]. The CT state, on the other hand, is most stabilized near the orthogonal geometry. In DP-Mes, strong mixing of ET into $S_1$ suppresses the tendency towards planarisation, resulting in a relatively small Stokes shift (30 meV) and efficient one-step SF since the molecule maintains orthogonality. In DP-TIPS, the weaker interaction between $S_1$ and ET allows more significant torsional relaxation in $S_1$ before CT is formed, giving the larger Stokes shift (60 meV, and significantly reduced in rigid polymer matrix, Fig. 2e). This picture agrees well with our calculations, which reveal larger forces for excited-state relaxation in the DP-TIPS singlet state and indicate a greater overall geometric change (Supplementary Fig. 25). We expect that this stage of the process is where solvent polarisability becomes significant. In 9,9′-bianthryl, it has been shown theoretically[43] and experimentally[44,45] that lowering of the inter-unit angle increases the polarisability of $S_1$, and we anticipate similar behaviour in DP-TIPS. The excess polarisability in relaxed $S_1$ causes a rapid rearrangement of solvent molecules which is more pronounced with increased solvent polarisability. Upon conversion from $S_1$ into CT, this relaxed solvent electronic cloud presents a barrier against a return to the orthogonal CT geometry where triplet-pair formation is most efficient. Hence the CT→TT rate generally decreases as solvent polarisability increases. These behaviours strongly indicate that SF is intimately related to molecular geometry and suggest it could be a vibronically driven process as in the canonical pentacene system[46–48]. This possibility is the subject of an ongoing study. It is interesting to consider the situation in rigid PS matrix, in which the interchromophore geometry and relevant energy levels are essentially fixed. When large-amplitude motion is restricted, SF nonetheless proceeds in a rapid one-step process in both dimers. In the case of DP-TIPS, this geometric restriction entails a transition from 'real' CT mediation to either a virtual CT process[13,15,49] or direct $S_1$-TT coupling[4,10]. We note the matrix elements for the latter mechanism are particularly unfavourable (the coupling should be null in a purely orthogonal structure). Given the unambiguous role of CT states in DP-TIPS in solution and in DP-Mes in all media, we consider SF is most likely mediated by virtual CT states.

The dimers we present here offer unique tuneability, both statically through side-group engineering or solvent environment and dynamically through changes in molecular geometry. These results not only provide important new insight into the mechanism of SF but also highlight the power of side-group engineering to fine-tune material properties for SF. Functional groups such as -Mes and -TIPS are typically used to improve processability and manipulate solid-state packing. Our work demonstrates that they can also have a profound effect on molecular photophysics and thus serve as a convenient tool to optimize SF energetics.

## Methods

**Dimer synthesis.** The precursor for DP-Mes and DP-TIPS was 6,6′-bispentacenequinone **1**, which was prepared using established procedures[50]. DP-Mes and DP-TIPS were synthesized according to established literature protocols[8,18]. Details of synthesis, NMR characterisation and X-ray crystallographic characterisation of P-Mes, DP-Mes and DP-TIPS can be found in previous publications[8,18]. P-TIPS was purchased from Sigma Aldrich and used as received.

**Sample preparation.** The solubilizing side groups on pentacene derivatives provide chemical stability and render significant solubility. We have utilized this property to study well-isolated molecules of DP-Mes and DP-TIPS in solution or dispersed in polymer matrices. Unless otherwise stated, all measurements in this work were performed on solutions at 0.5 mg ml$^{-1}$ in all solvents, prepared and sealed under nitrogen atmosphere. Triplet sensitisation was performed using established procedures[37], with a mixed toluene solution of 0.5 mg ml$^{-1}$ DP-Mes and 1.5 mg ml$^{-1}$ N-methylfulleropyrrolidine. All solution measurements were performed in 1 mm light-path quartz cuvettes (Hellma Analytics). To prepare films with dispersed DP-TIPS, stock solutions of 5 mg ml$^{-1}$ DP-TIPS and 100 mg ml$^{-1}$ polystyrene, polyethylene oxide or polyvinyl acetate in toluene were mixed together to obtain a final DP-TIPS concentration of 1 mg ml$^{-1}$ with a DP-TIPS:polymer weight ratio of 1:99. This mixture was drop-casted on Spectrosil quartz substrates in nitrogen atmosphere. To obtain samples that gave phosphorescence, DP-Mes or DP-TIPS was mixed with platinum octaethylporphyrin (PtOEP, purchased from Sigma Aldrich) and dispersed in polystyrene matrix. The final weight percentage of dimer: PtOEP: polymer in the mixture was 1:5:95. This mixture was drop-casted on Spectrosil.

**Photoluminescence spectroscopy.** The PL of DP-TIPS was measured in two distinct temporal regimes. Fast (sub-100-ps) photoluminescence dynamics were studied using the transient grating technique described in detail in Chen et al.[33] Longer-time dynamics were recorded with a standard time-correlated single-photon counting system (Edinburgh Instruments), using 40 MHz excitation at 470 nm (PicoQuant), and delayed photoluminescence spectra were collected with an intensified CCD (iStar DH740, Andor Instruments). Phosphorescence was detected using a calibrated infrared InGaAs photodiode array (ANDOR iDus 490A) coupled to a spectrograph (ANDOR Shamrock), with CW excitation at 532 nm (0.7 mW).

**Transient absorption spectroscopy.** Transient absorption measurements were performed on a previously reported setup[37]. Briefly, broad-band probe pulses were generated using noncollinear optical parametric amplifiers (NOPAs) built in-house to cover three separate spectral ranges: 500–800 nm, 800–1150 nm and 1175-1550 nm. The same InGaAs array detector (Hamamatsu G11608-512) was used for all wavelengths. For sub-picosecond resonant excitation, DP-Mes and DP-TIPS were pumped with the 620 nm or 640 nm output from an automated OPA (TOPAS, Light Conversion), with a pulse duration of <200 fs, unless otherwise mentioned. The sub-ps setup was limited by the length of the mechanical delay stage to delays of 2 ns. Further spectral evolution and triplet sensitisation were investigated using excitation with the ~1 ns output of a frequency-doubled (532 nm) Q-switched Nd-YVO$_4$ laser (Advanced Optical Technologies), which was externally triggered with an electronic pulse. For these measurements, strong pump scatter in the spectral range 520–540 nm required removal of this probe region. In all measurements, pump and probe polarisations were set to magic angle (54.7°). Typical excitation densities were $10^{14}$–$10^{15}$ photons per pulse cm$^{-2}$, and all decay kinetics were found to be independent of pump intensity.

**Triplet yield determination.** We determined the yield of triplet excitons in DP-Mes and DP-TIPS following our previously reported method[8]. Briefly, we determine the ratio of singlet and triplet oscillator strengths at a given wavelength through normalisation of the spectra at the GSB peak. Our approach is based on the following assumptions: (1) the initial singlet exciton is delocalized over the entire dimer molecule and thus bleaches both pentacene cores, as described previously;[8] (2) a single triplet exciton on the dimer also bleaches the entire ground-state singlet transition; (3) the overlap of the TA spectral shape of $S_1$ and $T_1$ in the GSB range 615–625 nm (DP-Mes) and 635–640 nm (DP-TIPS) is due to the absence of overlapping features (SE and PIA) in this region, allowing the $T_1$ spectrum to be normalized relative to $S_1$; and (4) the PIA of two triplet excitons on a dimer is twice that of a single triplet. Assumption 2 follows directly from the optical evidence of inter-chromophore coupling such as strongly reduced oscillator strength. Covalent bonding to a second pentacene strongly alters the absorption of the first, and it is reasonable to expect that anything that affects the absorption of one of these pentacenes (such as a localized triplet exciton) should accordingly alter the absorption of the entire molecule. Any such alteration of the absorption spectrum is precisely what is measured by GSB, which is insensitive to the nature of the excitations present. We note that this does not necessarily mean the second, 'unexcited' pentacene cannot absorb light, but that the 'dimer' singlet transition, which is responsible for the GSB signal, is fully bleached. From the similarity of TT and single-triplet (that is, triplet + ground-state partner) PIA spectra in the wavelength range 500–1600 nm, we infer that there must be a substantial energetic penalty for an additional singlet exciton to reside in such close proximity to the triplet. There is no signature of a singlet transition for the other half of the dimer in our sensitized triplet spectra, meaning any such transition must be shifted to below 500 nm. As for the third assumption, we note that the constant GSB we observe in that spectral region during SEF requires either that there are no overlapping features or that they precisely and coincidentally balance out, which we consider highly unlikely, particularly given that we observe the same behaviour in each dimer in different spectral regions. We thus treat the triplet sensitisation spectrum as having the same underlying GSB (that is, of the entire molecule) as the singlet spectrum, and can normalize T relative to $S_1$ to determine the triplet oscillator strength. The final assumption is a consequence of the nature of PIA, which should be sensitive to the nature and directly proportional to the number of excitons

present in the system. From the change in decay kinetics between triplets generated through sensitisation and direct excitation, it is evident that the dimer is capable of supporting two localized triplet excitons, and it stands to reason that each can be excited into higher-lying triplet states.

Following these assumptions, two parameters were needed to calculate the triplet yield: the ratio of singlet and triplet excited-state molar extinction coefficients ($\varepsilon^*$) and the ratio of the singlet and triplet excited state absorption under direct excitation. To determine the ratio of $\varepsilon^*$, we compared the PIA of $S_1$ at 100 fs (when the stimulated emission is maximum) and the PIA of $T_1$ obtained from sensitisation, scaled such that the GSB of each has the same magnitude over 615–625 nm for DP-Mes and 635–640 nm for DP-TIPS (see Supplementary Fig. 26 for normalized spectra). We then apply the formula below, using PIA magnitudes taken at 100 fs (initial $S_1$ population) and 10 ps ($T_1$ population following SEF) after direct excitation.

$$\Phi_{\text{Triplet}} = \frac{\Delta A(T_1)|_{t=10\,\text{ps}}}{\Delta A(S_1)|_{t=100\,\text{fs}}} \times \frac{\varepsilon^*_{S1}}{\varepsilon^*_{T1}} \qquad (1)$$

In solvents in which $S_{\text{stab}}$ or CT does not fully convert into triplets, the contribution of the residual population was first removed from the signal at 10 ps via spectral decomposition. Details of the calculations are presented in the Supplementary Note 2.

**Rate constant calculation.** The underlying rates of the processes described in this work—SF, singlet stabilisation, formation of CT states and stabilisation of CT—were determined from the time-constants and branching ratios measured in TA. Branching ratios for competing processes such as SF versus $S_{\text{stab}}$ formation were extracted from spectral decomposition kinetics on the assumption that the absorption cross-section of the stabilized state is unchanged from the initially formed state. These parameters were used in the following expressions:

$$K_{\text{total}} = \frac{1}{\tau_{\text{TA}}} = K_{\text{SF}} + K_{\text{Sstab}} \qquad (2)$$

$$\text{frac}_{\text{SF}} = \frac{K_{\text{SF}}}{K_{\text{tot}}} \text{ and } \text{frac}_{\text{Sstab}} = \frac{K_{\text{Sstab}}}{K_{\text{tot}}} \qquad (3)$$

Rates and branching ratios are tabulated in Supplementary Tables 2 and 3 and TA time constants in Supplementary Tables 4–9.

**Data availability.** The data underlying this publication are available at http://dx.doi.org/10.17863/CAM.5937.

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

## Acknowledgements

S.L. thanks AGS(O) Scholarship support from A*STAR Singapore. J.W. acknowledges financial support from MOE Tier 3 grant (MOE2014-T3-1-004). This work was supported by the Engineering and Physical Sciences Research Council, U.K. (Grant numbers EP/M005143/1 and EP/G060738/1). D.H.P.T. and N.D.M.H. acknowledge the Winton Programme for the Physics of Sustainability. K.C. and J.M.H. acknowledge support from a Rutherford Discovery Fellowship to J.M.H. A.J.M. gratefully acknowledges Jenny Clark for useful discussions.

## Author contributions

S.L. and A.J.M. planned experiments, S.L. carried out all experiments except ultrafast PL and performed data analysis, K.C. and J.M.H. performed ultrafast PL experiments, D.H.P.T. and N.D.M.H. performed calculations, S.D. and J.W. synthesized both dimers, all authors discussed results, and S.L., N.C.G. and A.J.M. wrote the manuscript.

## Additional information

**Competing financial interests:** The authors declare no competing financial interests.

