## [Peer Review File · Nature Communications]

Reviewers' comments:

Reviewer #1 (Remarks to the Author):

Summary: The manuscript of Musser provides an interesting investigation of the role of CT states on singlet fission in pentacene dimers, which is both timely and interesting, and is in my view publishable in Nature Communications.

Detailed Assessment:

A. Summary of the key results:

The manuscript from Musser and co-workers provides an investigation of singlet fission, and in particular the influence of solubilising groups and the role of the matrix (polarity, rigidity) on intramolecular singlet fission in pentacene dimers. They use the fact that the polarity of the matrix has a strong influence on the energetic structure of the molecules, and in particular can be used to systematically tune the energy level of an intermediate charge transfer (CT) state, to demonstrate the strong influence that this CT state has on SF. They further show that the rigidity of the matrix strongly influences SF, with rigid molecules limiting accessibility of the CT state and allowing direct SF.

B. Originality and interest: if not novel, please give references:

Whilst the techniques used here are not novel (there is for example a large literature regarding the influence of solvent polarity on CT state energetics) the application of this to investigate intramolecular SF is, as far as I know, new. It provides a level of insight into the SF process in these molecules that will be of wide interest to the SF community. It will be of use to those designing new molecular systems for SF, and those working to incorporate SF materials in to photovoltaic device architectures.

In my view this work is of a level of interest and accessibility that would make it suitable for publication in Nature Comms.

C. Data & methodology: validity of approach, quality of data, quality of presentation:

The data are obviously well taken, and of high quality. The data is reasonably well presented (although some improvement to the line thickness in figure 1 could help readability, and I personally find the color selection makes discriminating data challenging in the majority of the figures) .

The logical argument is easy to follow and sound.

D. Appropriate use of statistics and treatment of uncertainties:

The errors reported seem reasonable.

E. Conclusions: robustness, validity, reliability:

The conclusions presented in this manuscript appear robust and valid. The clear relation between polarizability and timescales seen across the many solvents used points to the robustness of the result.

F. Suggested improvements: experiments, data for possible revision:

I have no suggestions for changes.

G. References: appropriate credit to previous work?

Appropriate credit is given.

H. Clarity and context: lucidity of abstract/summary, appropriateness of abstract, introduction and conclusions:

The abstract introduction and conclusions clearly and succinctly describe the work presented.

Reviewer #2 (Remarks to the Author):

This is an excellent comprehensive study of the role of charge transfer states as an intermediate in intramolecular singlet fission observed by transient absorption spectroscopy. Singlet fission is currently a hot topic both in photophysical chemistry and also in organic photovoltaic community. Many theoretical models have proposed that charge transfer states play a crucial role in singlet fission. However, no clear spectroscopic evidence has been reported so far. In this study, charge transfer state is observed as an intermediate in singlet fission, which is assigned on the basis of solvent-polarity dependence. These experiments and analyses have been performed with care, and the results and discussion are consistent and understandable. As such, I suggest acceptance of this work in Nature Communications after the authors have considered the following issues.

1) Intramolecular singlet fission

In this study, triplet excitons formed by singlet fission should not be free triplets but triplet pairs because each pentacene dimer is isolated in solution. For the benefit of the reader, I think it would be better to abbreviate triplet pairs as (TT) rather than (T + T). In general, (T + T) represents free triplets.

2) Transient absorption spectra

As mentioned above, the solvent-polarity dependence they observed is strongly indicative of the contribution of the CT state to singlet fission. However, I feel it is difficult to distinguish the CT state from triplet pairs spectroscopically because both absorption spectra are very similar to each other in the wavelength range they observed as shown in Figure 4. As reported in JACS 2015, 137, 15980, it might be possible to distinguish between the two transient species spectroscopically by measuring transient absorption not only in the near-IR but also in the short-wavelength IR. I would suggest the authors perform additional experiments in order to support their key finding of the spectroscopic observation of the CT states. Such clear data would be beneficial for the broad audience other than photophysical chemists.

3) Superexchange-type mediation

On page 16, the authors insist the efficient singlet fission in DP-TIPS in rigid PS matrix is the most direct demonstration of the superexchange-type mechanism. They should have explained this logic more clearly. I do not understand how they rule out the possibility of the direct singlet fission.

4) Intermolecular singlet fission

I feel it would be beyond the scope of this paper to discuss the intermolecular singlet fission because the singlet fission in this study is basically intramolecular process even in rigid matrix.

Reviewer #3 (Remarks to the Author):

The report by Lukman et al. presents spectroscopic data (and some limited calculations) on two pentacene dimers that undergo singlet fission in solution. The manuscript is thorough in its description

and presentation of experimental results and is of high relevance to the field, although many of the observations were already reported by the same authors in ref 8 for one of the dimers (including some matrix and solvent effects). Still, there is some novelty in the data and the analysis. There are some nice features that show how the tuning of the CT energy impacts the route that SF can take. If some important points can be clarified and if the overall language can be made more precise, the manuscript could be published.

In the introduction but also in other places in the manuscript, earlier literature on dimers is not cited despite a fairly clear similarity between the concepts being considered, especially the effects of conformation and solvent on S1 and CT state energies. This includes some Michl dimer papers (DOI: 10.1021/jp310979q and DOI: 10.1021/acs.jpca.6b00826) as well as a Wasielewski paper (DOI: 10.1038/NCHEM.2589). I realize some of these are very new reports, but they should be mentioned to refer the reader to similar works on different molecular dimers.

I find it odd that Figure 1 is essentially a summary of the eventual conclusions but is shown at the beginning. I can understand the desire to have a schematic figure to better define the hypothesis, but the caption is written such that everything is already known. A more logical ordering would be to have the manuscript use data/theory to build evidence toward the ultimate schematic picture, not vice versa. Some portion of Fig 1 could remain the same, but more graphical information could be added, such as depicting the geometries of the various states, as suggested below. Then, a more detailed figure outlining the various schemes determined from the data could come toward the end.

For an audience that is not particularly specialized in singlet fission or the photophysics of dimers, it is imperative to provide the reader more information about the optimized geometries and the wavefunctions for at least some of the structures Sstab, S*, CT, T1, and TT. There is much discussion of conversion between these states, the importance of the interchromophore geometries, the relatively degrees of delocalization as opposed to localization...but, with no actual picture of the optimized geometries and wavefunctions of these states, it is difficult to get a clear idea of the photophysics. Only the planarized versions of the dimers are shown as a small part of Figure 1. I realize that generating realistic figures based on calculations could be a large undertaking, but it is worthwhile both for making the manuscript more comprehensible and for supporting the conclusions.

The other major issue is with regard to assumption (2) in the "Triplet Yield Determination" section, which is also used in the caption of Figure 4. It is strange to see the verbose discussion of this in the "experimental methods" section, since most of the language is highly conceptual. In assumption (1) it is stated that the reason for singlet excitons causing a full bleach of the dimer is that such excitations are entirely delocalized over both cores. Some justification for the lack of change in absorption spectrum despite the delocalization, concerning the cancellation of excitonic and dark CT effects, is at least partially convincing, although the striking similarity between spectra still leads to some question about truly how extensive the delocalization can be. These are fairly large systems, but could some calculations be done to justify the claim that this dimer in particular can support such excitations, and that the relevant energies of the states compared with the monomers, remain essentially unchanged? For assumption (2), involving triplet states, the complete bleach of the ground state is said to be happening for the same reason. What reason? Complete delocalization? Later, the triplet excitons are said to be "localized", which one would expect. If this is not a direct contradiction, at least it is confusing.

It is later stated that anything happening on the neighboring chromophore, such as production of a localized triplet exciton, will alter the ground state absorption of the other core. I understand that this may be true, but will it render the chromophore completely dark? I am struggling to understand how the triplet can completely block the production of a singlet exciton on the neighboring core yet the

molecule can easily support two triplet excitons without any energetic penalty or perturbation of the triplet spectra. This may not be the main theme of the paper, but it is of fundamental interest, and some of the conclusions do hinge on the veracity of this claim.

Some relatively minor comments:

The introduction cites published work establishing a correlation between SF rate and solvent polarity. Another paper in the literature (ref 30) suggests no or weak solvent dependence on solvent polarity for similar dimers, but for some reason is ignored in this important context in the introduction.

The statement: "modify the separation between CT states"? – maybe it is intended to be "modifies the separation between CT and LE states"?

The authors state: "...first detailed spectroscopy study of the solvent dependence of SF" – this is arguable and nonetheless a fairly vacuous statement. There are other studies of solvent dependence – what is considered "detailed"? Some other "firsts" mentioned throughout the text are equally dubious – especially considering the recent Wasielewski Nature Chem. paper.

The authors state: "proximity of singlet and CT states". CT states can also be of singlet or triplet spin, so the phrasing is unclear.

The authors state: "direct interaction between the two pentacene units" – direct as opposed to "indirect"? This is unclear phrasing.

The emissive CT state is stated to be fully a localized electron on one pentacene unit and a localized hole on the other? Shouldn't such a species have zero oscillator strength for radiative decay?

In the text it is mentioned that there is no aggregation in any solvent, but then Figure S2 shows that there is aggregation under some conditions. Also, it is said that there is no concentration dependence to the TTA rates, but no data are shown.

Another example of improper phrasing: "Very non-polar". Is it even more non-polar than just "non-polar"?

The reference to SF as a "vibronically-driven process" in this particular case is spurious. The motions discussed here are torsional, not vibrational, and include solvent degrees of freedom. This regime is very far from that considered in refs 39 and 41.

The authors state: "Interestingly, the rate constant displays a good correlation to solvent polarizability (Fig. 5b)." At best, this is a weak correlation.

If I understand correctly, relaxation of S1 into Sstab (by about 0.1 eV according to Fig 1) results in a situation strongly inhibited from undergoing SF. However, Sstab emits readily with a spectrum similar to S1, but slightly red-shifted. Thus, one should be able to directly excite Sstab at lower photon energies and produce essentially zero triplet yield. In other words, the action spectrum for triplets should not be constant across the absorption profile of the dimer. The allowed S0-Sstab transition should be stronger than the ground state to CT state transition that was shown to be fairly easy to observe. If not, is there a geometrical barrier? Some depiction of the involved states and the associated potential energy surfaces could be helpful.

REVIEWERS' COMMENTS:

Reviewer #1 (Remarks to the Author):

I am satisfied with the response of the authors and, based on my previous assessment of the novelty and interest of this work, recommend that it be published in Nature Communications.

Reviewer #2 (Remarks to the Author):

In response to the previous comments, the authors have revised their manuscript appropriately. I am therefore pleased to recommend acceptance in Nature Communications. No further review is needed.

Reviewer #3 (Remarks to the Author):

The revised manuscript takes into account my suggested modifications in a satisfactory way. I thank the authors for taking these suggestions seriously. The excitation wavelength dependent triplet yield is a particularly nice addition.

My only comment to the authors is to read the revised sections for minor errors, particularly the SI. For example, the symbol "@" is now being used in some places instead of the word "at", but not in any consistent way. Also, some simplification of the figures might be helpful. The purpose of the small potential energy surface shown in Figure 1 is not clear. It probably should be removed. Finally, the "thick" vs. "thin" lines in Fig 8 that designate efficient vs. inefficient processes are fairly difficult to discern. You might consider a different designation.

We are grateful to the reviewers for their detailed reading of our manuscript and strong endorsement of our work. We consider that we have been able to fully address all of the reviewers' main concerns, which we detail point by point below. We also provide a revised version of the main text and supplementary information with all changes highlighted yellow to aid evaluation.

The primary changes can be summarised as follows:

1. Reformatting of figures to improve readability.
2. Design of a new, more generalised scheme for Figure 1, with the original detailed version moved to the discussion section.
3. Further calculations supporting and discussion regarding the inter-pentacene coupling in the dimers, including a new figure showing the wavefunctions for two states (Figure 3).
4. New transient absorption measurements with a broader probe range and a detailed pump wavelength excitation series to better characterise the CT state and explore possible direct excitation of S_{stab} and CT.

Reviewers' comments:

Reviewer #1 (Remarks to the Author):

Summary: The manuscript of Musser provides an interesting investigation of the role of CT states on singlet fission in pentacene dimers, which is both timely and interesting, and is in my view publishable in Nature Communications.

Detailed Assessment:

A. Summary of the key results:

The manuscript from Musser and co-workers provides an investigation of singlet fission, and in particular the influence of solubilising groups and the role of the matrix (polarity, rigidity) on intramolecular singlet fission in pentacene dimers. They use the fact that the polarity of the matrix has a strong influence on the energetic structure of the molecules, and in particular can be used to systematically tune the energy level of an intermediate charge transfer (CT) state, to demonstrate the strong influence that this CT state has on SF. They further show that the rigidity of the matrix strongly influences SF, with rigid molecules limiting accessibility of the CT state and allowing direct SF.

B. Originality and interest: if not novel, please give references:

Whilst the techniques used here are not novel (there is for example a large literature regarding the influence of solvent polarity on CT state energetics) the application of this to investigate intramolecular Sf is, as far as I know, new. It provides a level of insight into the SF process in these molecules that will be of wide interest to the SF community. It will be of use to those designing new molecular systems for SF, and those working to incorporate Sf materials in to photovoltaic device architectures.

In my view this work is of a level of interest and accessibility that would make it suitable for publication in Nature Comms.

C. Data & methodology: validity of approach, quality of data, quality of presentation:

The data are obviously well taken, and of high quality. The data is reasonably well presented (although some improvement to the line thickness in figure 1 could help readability, and I personally find the color selection makes discriminating data challenging in the majority of the figures).

We thank the reviewer for this positive assessment. We have reformatted the original Figure 1 (now Figure 8, following the suggestion of Reviewer 3 below) and changed the colour schemes in Figures 1, 5, 6 and 7 to aid visibility.

The logical argument is easy to follow and sound.

D. Appropriate use of statistics and treatment of uncertainties:

The errors reported seem reasonable.

E. Conclusions: robustness, validity, reliability:

The conclusions presented in this manuscript appear robust and valid. The clear relation between polarizability and timescales seen across the many solvents used points to the robustness of the result.

F. Suggested improvements: experiments, data for possible revision:

I have no suggestions for changes.

G. References: appropriate credit to previous work?

Appropriate credit is given.

H. Clarity and context: lucidity of abstract/summary, appropriateness of abstract, introduction and conclusions:
The abstract introduction and conclusions clearly and succinctly describe the work presented.

Reviewer #2 (Remarks to the Author):

This is an excellent comprehensive study of the role of charge transfer states as an intermediate in intramolecular singlet fission observed by transient absorption spectroscopy. Singlet fission is currently a hot topic both in photophysical chemistry and also in organic photovoltaic community. Many theoretical models have proposed that charge transfer states play a crucial role in singlet fission. However, no clear spectroscopic evidence has been reported so far. In this study, charge transfer state is observed as an intermediate in singlet fission, which is assigned on the basis of solvent-polarity dependence. These experiments and analyses have been performed with care, and the results and discussion are consistent and understandable. As such, I suggest acceptance of this work in Nature Communications after the authors have considered the following issues.

We thank the reviewer for the strong feedback about the quality and relevance of our study and its presentation.

1) Intramolecular singlet fission: In this study, triplet excitons formed by singlet fission should not be free triplets but triplet pairs because each pentacene dimer is isolated in solution. For the benefit of the reader, I think it would be better to abbreviate triplet pairs as (TT) rather than (T + T). In general, (T + T) represents free triplets.

The reviewer makes a good point that the triplets formed via SF in our systems are tightly confined within the dimer. While we have no direct evidence of interaction between the triplets as in the correlated triplet pair reported in TIPS-tetracene solutions (Stern et al., PNAS 2015) or pentacene dimers reported by the Campos group (Sanders et al., JACS 2015), they are indeed not 'free'. We have thus changed the notation throughout from T+T to TT.

2) Transient absorption spectra: As mentioned above, the solvent-polarity dependence they observed is strongly indicative of the contribution of the CT state to singlet fission. However, I feel it is difficult to distinguish the CT state from triplet pairs spectroscopically because both absorption spectra are very similar to each other in the wavelength range they observed as shown in Figure 4. As reported in JACS 2015, 137, 15980, it might be possible to distinguish between the two transient species spectroscopically by measuring transient absorption not only in the near-IR but also in the short-wavelength IR. I would suggest the authors perform additional experiments in order to support their key finding of the spectroscopic observation of the CT states. Such clear data would be beneficial for the broad audience other than photophysical chemists.

The reviewer is right that additional spectral bandwidth is helpful to distinguish these otherwise-similar excited states. We have performed additional TA measurements on DP-TIPS over the probe range 1100-1500 nm for a wide range of solvents and polymer matrices. We have found that the CT state has a pronounced PIA ~1300 nm due to the cation component, while no such peak is observed in the triplet or TT states. These results enable a clearer distinction between the excited-state species and have been incorporated into the relevant figures (Figs. 5, 7, S11, S13, S14, S18 and S25). We also performed measurements in this spectral range for DP-Mes, but these yielded no useful additional information beyond what is evident in the original spectral range. We present some of those measurements in Fig. S24, but leave the original spectral range elsewhere to aid visibility of the primary spectral features of S1, TT and S*.

3) Superexchange-type mediation: On page 16, the authors insist the efficient singlet fission in DP-TIPS in rigid PS matrix is the most direct demonstration of the superexchange-type mechanism. They should have explained this logic more clearly. I do not understand how they rule out the possibility of the direct singlet fission.

We apologise that the logic was somewhat unclear in this statement. We consider the direct mechanism unlikely to be relevant here because 1) the coupling between S1 and TT should be vanishingly small in the equilibrium orthogonal geometry, and 2) the strong role of CT in all other situations in both dimers suggests it has suitable energetics and couplings to mediate the process even in a 'virtual' role. The new discussion now reads as follows:

(p18) 'It is also interesting to consider the situation of DP-Mes or -TIPS embedded in rigid PS matrix, in which the interchromophore geometry and relevant energy levels are essentially fixed. When large-amplitude motion is restricted, SF nonetheless proceeds in a rapid one-step process in both dimers. In the case of DP-TIPS, this simple geometric restriction entails a transition from 'real' CT mediation to either a virtual CT process^{13,15,53} or direct S₁-TT coupling^{4,10}. We note the matrix elements for the latter mechanism are particularly unfavourable (the coupling should be null in a purely orthogonal structure). Given the unambiguous role of CT states in DP-TIPS in solution and in DP-Mes in all media, we consider it most likely that SF is mediated by virtual CT states.'

4) Intermolecular singlet fission: I feel it would be beyond the scope of this paper to discuss the intermolecular singlet fission because the singlet fission in this study is basically intramolecular process even in rigid matrix.

The reviewer is correct that this aspect of the discussion is beyond the scope of our study of isolated dimer molecules. We have removed the direct comparison to intermolecular SF in the final paragraph.

Reviewer #3 (Remarks to the Author):

The report by Lukman et al. presents spectroscopic data (and some limited calculations) on two pentacene dimers that undergo singlet fission in solution. The manuscript is thorough in its description and presentation of experimental results and is of high relevance to the field, although many of the observations were already reported by the same authors in ref 8 for one of the dimers (including some matrix and solvent effects). Still, there is some novelty in the data and the analysis. There are some nice features that show how the tuning of the CT energy impacts the route that SF can take. If some important points can be clarified and if the overall language can be made more precise, the manuscript could be published.

We thank the reviewer for recognizing the relevance and quality of our study and our novel elucidation of the role of tuning the CT state. We believe that the changes detailed below are sufficient to address the points raised.

In the introduction but also in other places in the manuscript, earlier literature on dimers is not cited despite a fairly clear similarity between the concepts being considered, especially the effects of conformation and solvent on S₁ and CT state energies. This includes some Michl dimer papers (DOI: 10.1021/jp310979q and DOI: 10.1021/acs.jpca.6b00826) as well as a Wasielewski paper (DOI: 10.1038/NCHEM.2589). I realize some of these are very new reports, but they should be mentioned to refer the reader to similar works on different molecular dimers.

The reviewer is correct that the Michl dimer papers on diphenyl-isobenzofuran have some relevance to the CT-mediated mechanism for intramolecular SF. We had intended to focus solely on acene dimers, but agree that it would be appropriate to cite this earlier work. We have now incorporated a relevant citation in the introductory section. As for the Wasielewski paper, we note that work was accepted after our submission date and we thus could not have cited it in our original draft (indeed were not aware of the full conclusions of the study). Given its relevance to and complementarity with our results, we have also cited that work, noting however the important distinction that in the terrylene dimers the CT state is only directly observed in conditions in which SF is inefficient. Finally, we acknowledge the recent study of the Campos group in which no significant role was found for CT in SF in a pentacene dimer, highlighting the importance of building a clearer picture of when and how CT states contribute. The relevant section now reads:

(p2) These CT states are difficult to observe directly, but evidence of a solvent-polarity-dependent SF rate in two classes of pentacene dimer,^{8,11,12} and dimers of 1,3-diphenylisobenzofuran¹⁶ support such a model. We note that a study of solvent-dependent SF in terrylene diimide dimers¹⁷ was also very recently released, showing a direct competition between the formation of CT states and of triplet pairs. At the same time, some efficient SF dimers exhibit relatively little solvent dependence and appear to follow a direct S₁→TT mechanism,¹⁰ raising the need for further systematic investigation of the nature and role of CT states in SF

I find it odd that Figure 1 is essentially a summary of the eventual conclusions but is shown at the beginning. I can understand the desire to have a schematic figure to better define the hypothesis, but the caption is written such that everything is already known. A more logical ordering would be to have the manuscript use data/theory to build evidence toward the ultimate schematic picture, not vice versa. Some portion of Fig 1 could remain the same, but more graphical information could be added, such as depicting the geometries of the various states, as suggested below. Then, a more detailed figure outlining the various schemes determined from the data could come toward the end. For an audience that is not particularly specialized in singlet fission or the photophysics of dimers, it is imperative to provide the reader more information about the optimized geometries and the wavefunctions for at least some of the structures Sstab, S*, CT, T1, and TT. There is much discussion of conversion between these states, the importance of the interchromophore geometries, the relatively degrees of delocalization as opposed to localization...but, with no actual picture of the optimized geometries and wavefunctions of these states, it is difficult to get a clear idea of the photophysics. Only the planarized versions of the dimers are shown as a small part of Figure 1. I realize that generating realistic figures based on calculations could be a large undertaking, but it is worthwhile both for making the manuscript more comprehensible and for supporting the conclusions.

We recognize it was unusual to put such comprehensive results and conclusions into our initial schematic figure. The original Figure 1 has now been moved to the end of the paper as Figure 8, while a new, more basic schematic has been inserted as Figure 1. We have provided a very basic 'cartoon-level' indication of the expected molecular geometry at the relevant excited states. From comparison to literature (particularly that for bianthryl) and our spectroscopic measurements, we are confident that our depiction of the key coordinate (torsion about the interpentacene bond) is qualitatively correct: namely, the excitonic singlet state exhibits a tendency towards planarization, while the CT state is most stabilised in an orthogonal geometry. Likewise, the TT state evidently favours an orthogonal geometry, given the high SF rate in rigid matrices. The S^* state we deduce must entail some large-scale conformational relaxation, as it is completely suppressed in high viscosity or rigid matrix. A full determination of the potential energy surface or optimization of the excited state geometries is beyond the scope of this work and would have questionable accuracy using our methods. We have, however, computed singlet transition densities and electron-transfer state electron-hole densities, which are now presented in Figure 3, to give a better picture of the absorbing states (all are computed in the ground-state geometry, for vertical excitation). These calculations are supplemented with a new analysis of the excited-state forces in S_1 , which indicate a greater degree of geometric relaxation in the -TIPS dimer than in -Mes. These results are presented in Figure S27 and referred to near the end of the main text:

(p3) 'In order to model the vertical S_1 excitations of both dimers and their corresponding monomers, we performed time-dependent DFT (TDDFT) calculations using the NWChem code.¹⁹ The excitation energies and transition densities are summarized in Fig. 3a. We also performed constrained DFT calculations with ONETEP to model the pure electron transfer (ET) excitations.²⁰⁻²² Energies and electron-hole density plots are shown in Fig. 3b for the vacuum case, results with implicit solvent can be found in the Supplementary Information.'

(p12) 'In DP-TIPS, the weaker interaction between S_1 and ET allows more significant torsional relaxation in S_1 before CT is formed, giving rise to the larger Stokes shift (60 meV) in that molecule (we recall that the Stokes shift is significantly reduced in rigid polymer matrix, Fig. 2e). This picture agrees well with our calculations, which reveal larger forces for excited-state relaxation in the DP-TIPS singlet state and indicate a greater overall geometric change (Fig. S27).'

The other major issue is with regard to assumption (2) in the "Triplet Yield Determination" section, which is also used in the caption of Figure 4. It is strange to see the verbose discussion of this in the "experimental methods" section, since most of the language is highly conceptual.

We have put this discussion of our triplet yield determination in the Methods section as it was already reported in our initial dimer study, including our list of underlying assumptions. Moreover, we take the view that the importance of our results stems from the observed kinetics and the nature of the excited states involved. Though we have taken great care to ensure our triplet yield determination is robust and well defended, the primary conclusions of our study are not strongly affected by the resulting values, hence we have relegated the discussion of the technique to this supplementary section.

In assumption (1) it is stated that the reason for singlet excitons causing a full bleach of the dimer is that such excitations are entirely delocalized over both cores. Some justification for the lack of change in absorption spectrum despite the delocalization, concerning the cancellation of excitonic and dark CT effects, is at least partially convincing, although the striking similarity between spectra still leads to some question about truly how extensive the delocalization can be. These are fairly large systems, but could some calculations be done to justify the claim that this dimer in particular can support such excitations, and that the relevant energies of the states compared with the monomers, remain essentially unchanged?

We agree that the delocalisation result is a surprising one, and the similarity between the spectra is puzzling. We can find no other alternative explanation for all of our experimental findings. For instance, the rapid intramolecular TTA we observe (<1 ns vs several us lifetime in sensitisation) requires that there be two triplets formed per molecule, but there is no corresponding growth of the GSB. Rather than rely on such arguments, we have tried to justify the claim with reference to exciton coupling theory. We have also performed further DFT calculations, revealing a Davydov split for symmetric and antisymmetric combinations of the monomer transition dipole moments: the lower state is bright and observed in our experiments, while the upper state is fully dark. We consider that the agreement of

these theoretical results with our data is a strong additional support for our interpretation of a delocalised singlet state. The relevant discussion in the main text has been changed as follows:

(p3-4) 'The spectra are also slightly broadened and shifted from the monomeric pentacenes: to the red in the case of DP-Mes and to the blue in DP-TIPS. The overall shift is the result of competing effects between delocalisation into side groups, coupling via dark CT states and dipole-dipole interactions between the two pentacenes.'

In order to model the vertical S_1 excitations of both dimers and their corresponding monomers, we performed time-dependent DFT (TDDFT) calculations using the NWChem code.¹⁹ The excitation energies and transition densities are summarized in Fig. 3a. We also performed constrained DFT calculations with ONETEP to model the pure electron transfer (ET) excitations.²⁰⁻²² Energies and electron-hole density plots are shown in Fig. 3b for the vacuum case, results with implicit solvent can be found in the Supplementary Information. As can be seen in Fig. 3a the -TIPS side-groups (specifically their C-C triple bonds) participate more strongly in the S_1 excitation than the -Mes groups. This factor is a major contributor to the lower S_1 energies of the TIPS molecules (compared to the Mes case), in accordance with experiment. The dimer S_1 states can essentially be interpreted as symmetric linear combinations of the corresponding monomer states. It is well known that the monomer transition dipoles are polarised along the short pentacene axis^{23,24}, i.e. along the direction of the connecting bond in the dimers. In the dimer S_1 state these monomer transition dipoles add constructively, producing a bright transition. In both dimers this bright state is accompanied by a corresponding dark state where the monomer dipoles are anti-parallel and cancel each other. This state is calculated to be 0.13 eV and 0.15 eV higher in energy for DP-Mes and DP-TIPS, respectively. This constitutes a significant exciton splitting with the interesting feature that the upper component is fully dark and therefore not detectable in absorption experiments. The calculations qualitatively reproduce the shifts observed when going from the monomeric pentacene to the respective dimer: to the red for DP-Mes and to the blue for DP-TIPS. We offer an interpretation of this behavior in terms of a competition between side-group and exciton-splitting effects. From monomer to dimer the ratio of side groups to pentacenes is reduced from 2:1 to 1:1. Since participation of the side groups in the excitation reduces the overall energy, this reduction is expected to result in a blue-shift which is particularly strong in DP-TIPS. This blue-shift competes with the red-shift due to coupling of transition dipoles which is similar in both molecules (approximately equal exciton splitting). In total this results in an overall red-shift for DP-Mes and a small blue-shift for DP-TIPS. Together these theoretical results support the notion of significant dipole-dipole coupling between the individual pentacenes upon dimerisation, in a way that is consistent with the observed shifts. This coupling energetically favours S_1 excitations which are fully delocalised over the dimers, in good agreement with the experimental data (see below). This should be the case even in the presence of some disorder, due to the robustness of the alignment of the dipoles associated with local excitations in each monomer.'

For assumption (2), involving triplet states, the complete bleach of the ground state is said to be happening for the same reason. What reason? Complete delocalization? Later, the triplet excitons are said to be "localized", which one would expect. If this is not a direct contradiction, at least it is confusing. It is later stated that anything happening on the neighboring chromophore, such as production of a localized triplet exciton, will alter the ground state absorption of the other core. I understand that this may be true, but will it render the chromophore completely dark? I am struggling to understand how the triplet can completely block the production of a singlet exciton on the neighboring core yet the molecule can easily support two triplet excitons without any energetic penalty or perturbation of the triplet spectra. This may not be the main theme of the paper, but it is of fundamental interest, and some of the conclusions do hinge on the veracity of this claim.

We recognise our language in this section was at times imprecise and may have been confusing. We have removed the unnecessary phrase 'for the same reason' and added further justification for assumption (2) further below. We also agree with the reviewer that it is in principle quite surprising that there is such strong agreement between the single-triplet sensitisation spectrum and that of the TT state – aside from the relative weight of PIA to GSB, there are no major differences. This is in contrast to some of the dimer reported by the Campos groups, where a distinct 'pair' signature is evident for dimers with short spacers. We consider it most likely that this behaviour in DP-Mes and DP-TIPS is due to the orthogonal geometry. In such a configuration the localised triplet wavefunctions should have negligible overlap, nor should they be able to interact via the dipole-dipole mechanism available to the singlet. It is important to recall in this context that the exact energy of the TT state is not known, as

there are very few systems where it has been possible to characterise it. In our schemes such as Fig. 7 we approximate the TT energy as $2xT_1$, which we have found to be sufficient for explanation of our results. The reworded section regarding triplet yield assumption (2) now reads:

(p15) 'Assumption 2 follows directly from the optical evidence of inter-chromophore coupling such as strongly reduced oscillator strength. Covalent bonding to a second pentacene strongly alters the absorption of the first, and it is reasonable to expect that anything that affects the absorption of one of these pentacenes (such as a localised triplet exciton) should accordingly alter the absorption of the entire molecule. Any such alteration of the absorption spectrum is precisely what is measured by GSB, which is insensitive to the nature of the excitations present. We note that this does not necessarily mean the second, 'unexcited' pentacene cannot absorb light, but that the 'dimer' singlet transition, which is responsible for the GSB signal, is fully bleached. From the similarity of TT and single-triplet (i.e. triplet + ground-state partner) PIA spectra in the wavelength range 500-1600 nm, we infer that there must be a substantial energetic penalty for an additional singlet exciton to reside in such close proximity to the triplet. There is no signature of a singlet transition for the other half of the dimer in our sensitised triplet spectra, meaning any such transition must be shifted to below 500 nm.'

Some relatively minor comments:

The introduction cites published work establishing a correlation between SF rate and solvent polarity. Another paper in the literature (ref 30) suggests no or weak solvent dependence on solvent polarity for similar dimers, but for some reason is ignored in this important context in the introduction.

The reviewer is correct, and this has been rectified. See comment above regarding Michl and Wasielewski papers.

The statement: "modify the separation between CT states"? – maybe it is intended to be "modifies the separation between CT and LE states"?

The phrase has been changed: (p3) 'which modifies the energetic separation between CT and vertical locally excited (LE) states'

The authors state: "...first detailed spectroscopy study of the solvent dependence of SF" – this is arguable and nonetheless a fairly vacuous statement. There are other studies of solvent dependence – what is considered "detailed"? Some other "firsts" mentioned throughout the text are equally dubious – especially considering the recent Wasielewski Nature Chem. paper.

We have removed most of these claims of 'first' as per the reviewer's suggestion. However, we maintain there is an important distinction between our work and the Wasielewski paper (which we reiterate had not been published at the time of our submission): where they observe efficient SF, there is no observation of CT, and the CT state must thus play at most a virtual role. By contrast, in DP-TIPS we have clear evidence of a directly populated CT state that subsequently undergoes rapid singlet fission. We have thus retained the phrasing on p3 with slight alteration:

'This is the first direct observation of a CT state intermediate to efficient SF'

The authors state: "proximity of singlet and CT states". CT states can also be of singlet or triplet spin, so the phrasing is unclear.

This has been altered: (p3) 'In DP-Mes, the proximity of S_1 and CT states results in strong mixing...'

The authors state: "direct interaction between the two pentacene units" – direct as opposed to "indirect"? This is unclear phrasing.

We have removed the unnecessary 'direct'.

The emissive CT state is stated to be fully a localized electron on one pentacene unit and a localized hole on the other? Shouldn't such a species have zero oscillator strength for radiative decay?

The reviewer is correct, and our original phrasing suggested that we observed a purely 'electron-transfer' type of CT state. However, the charge transfer is just the predominant character of the adiabatic state, and there must indeed be sufficient orbital overlap for weak radiative decay. We have reworded our descriptions of the state as follows:

(p6) 'We show below that this state resembles a radical anion localised on one pentacene and a radical cation on the other, though there must be sufficient wavefunction overlap between these to enable weak emission. This state is intermediate to efficient SF in DP-TIPS.'

(p9) 'This is a direct and unambiguous observation of a populated CT intermediate in an efficient SF system. The intramolecular CT state observed here is not a pure, diabatic electron-transfer state, but rather a partial symmetry-breaking charge separation between the dimer, as has been suggested in 9,9-bianthryl²⁹ and other systems^{17,45}. This is the reason it has non-zero oscillator strength, detected both through photoluminescence and sub-gap excitation.'

In the text it is mentioned that there is no aggregation in any solvent, but then Figure S2 shows that there is aggregation under some conditions. Also, it is said that there is no concentration dependence to the TTA rates, but no data are shown.

We have clarified that statement on p5 to clarify that no signatures of aggregation were detected in the solutions we used for spectroscopic study. The reviewer is correct that aggregation could be induced through particular solvent mixtures. We have also performed additional concentration-dependent measurements on DP-TIPS in polar and non-polar solvent, which are now presented in Supplementary Figs. S13-14. The sentence on p5 now reads:

'We note that there is no evidence in the absorption spectra of aggregation at the concentrations used, in any solvent, though aggregation can be induced in particular solvent mixtures (Supplementary Fig. S2).'

Another example of improper phrasing: "Very non-polar". Is it even more non-polar than just "non-polar"?

This has been removed.

The reference to SF as a "vibronically-driven process" in this particular case is spurious. The motions discussed here are torsional, not vibrational, and include solvent degrees of freedom. This regime is very far from that considered in refs 39 and 41.

We recognise the reviewer's concern that the two regimes being compared are quite different. Our intention with this comparison was to highlight the importance of molecular and interchromophore geometry, which clearly plays a direct role both in the classic systems and in our dimers. We are indeed currently pursuing further measurements using the techniques of refs 50 and 52 (orig. 39 and 41) to better understand if and how the reported vibronic coupling translates to these slower, torsionally controlled systems. We have altered the sentence on p13 to make it clear we are not currently claiming the same mechanism applies, but merely highlighting the possible connection:

'These behaviours strongly indicate that SF in both materials is intimately related to molecular geometry and suggest it could be a vibronically driven process as in the canonical pentacene system.⁵⁰⁻⁵² This possibility is the subject of an ongoing study.'

The authors state: "Interestingly, the rate constant displays a good correlation to solvent polarizability (Fig. 5b)." At best, this is a weak correlation.

We have removed the word 'good'.

If I understand correctly, relaxation of S1 into Sstab (by about 0.1 eV according to Fig 1) results in a situation strongly inhibited from undergoing SF. However, Sstab emits readily with a spectrum similar to S1, but slightly red-shifted. Thus, one should be able to directly excite Sstab at lower photon energies and produce essentially zero triplet yield. In other words, the action spectrum for triplets should not be constant across the absorption profile of the dimer. The allowed S0-Sstab transition should be stronger than the ground state to CT state transition that was shown to be fairly easy to observe. If not, is there a geometrical barrier? Some depiction of the involved states and the associated potential energy surfaces could be helpful.

Following the reviewer's suggestion, we have performed additional sub-gap excitation experiments, with a detailed excitation series of DP-Mes and DP-TIPS now presented in Supplementary Figs. S24-25. In DP-Mes the situation is relatively simple. For sufficiently low excitation energy, we do indeed directly excite long-lived S_{stab} and observe no triplet formation, with the threshold for >100% triplet formation roughly coinciding with the 0-0 peak energy of S_{stab} (due to overlap with the tail of S₁). We can also directly excite S* at slightly lower excitation energy, suggesting that torsional disorder at room temperature is sufficient to afford a small population of relatively planar molecules in the ground state. In DP-TIPS the sub-gap excitation behaviour is more complex, as S_{stab} does not relax from S₁ to as great a degree and there is considerable overlap with the CT state, which is generally lower in energy. What we observe in that molecule is increasing direct excitation of the CT state with progressively reduced excitation energy. At no point can we distinguish unique direct excitation of S_{stab}, which we expect is due to the closely overlapped energy ranges and geometric factors: the orthogonal GS geometry is more suited to CT than the relaxed singlet state. Interestingly, below a threshold ~coincident with the peak CT emission energy (in hexane), the triplet yield from CT is sharply reduced. This suggests that 'cold' CT states, like S_{stab} in DP-Mes, can be directly excited and are unable to undergo SF. We refer to these results in the main text as follows:

(p8) 'The excited state S_{stab} can be directly generated from sub-band gap excitation (Fig. S24e). Progressive reduction of the excitation energy results in a reduced triplet yield in favour of long-lived S_{stab} as SF becomes more and more endothermic (Fig. S24). At very low excitation energy (670 nm in hexane), a small population of S* can be directly excited (Fig. S24f), a consequence of room-temperature structural dynamics which yield a small population of relatively planarised dimers.'

(p10) 'The formation of TT can be suppressed through sub-gap excitation by directly exciting to lower CT energy levels available in the solvent range. Scanning the excitation wavelength for DP-TIPS through the polarity series (Fig. S25), we observe that beyond the threshold of 660 nm we increasingly directly excite CT rather than S₁. The triplet yield stays constant up to 680 nm ($\Delta E_{CT-TT} \sim 50$ meV) and decreases sharply for longer wavelengths due to the stabilisation of CT in higher polarity solvent.'

We are grateful to the reviewers for their detailed reading of our manuscript and strong endorsement of our work. We also provide a revised version of the main text and supplementary information.

Reviewer #1 (Remarks to the Author):

I am satisfied with the response of the authors and, based on my previous assessment of the novelty and interest of this work, recommend that it be published in Nature Communications.

We thank the reviewer for this positive assessment and recommendation.

Reviewer #2 (Remarks to the Author):

In response to the previous comments, the authors have revised their manuscript appropriately. I am therefore pleased to recommend acceptance in Nature Communications. No further review is needed.

We thank the reviewer for the positive feedback about the revision and recommendation to publish.

Reviewer #3 (Remarks to the Author):

The revised manuscript takes into account my suggested modifications in a satisfactory way. I thank the authors for taking these suggestions seriously. The excitation wavelength dependent triplet yield is a particularly nice addition.

My only comment to the authors is to read the revised sections for minor errors, particularly the SI. For example, the symbol "@" is now being used in some places instead of the word "at", but not in any consistent way. Also, some simplification of the figures might be helpful. The purpose of the small potential energy surface shown in Figure 1 is not clear. It probably should be removed. Finally, the "thick" vs. "thin" lines in Fig 8 that designate efficient vs. inefficient processes are fairly difficult to discern. You might consider a different designation.

We are very grateful for the comments, which have required us to improve the quality of our manuscript on several very important points. We have carefully reformatted the SI figures to ensure a uniform presentation, removed the potential energy surface from Figure 1 and emphasized the differences between 'thick' and 'thin' lines in the final Figure. Following the reviewer's comments, we have also elected to introduce a new main-text figure (Fig . 7) showing the excitation wavelength dependent triplet yield.